# Rapid attribution analysis of the extraordinary heatwave on the Pacific Coast of the US and Canada June 2021

Sjoukje Y. Philip[1,*], Sarah F. Kew[1,*], Geert Jan van Oldenborgh[1,2,†], Faron S. Anslow[3], Sonia I. Seneviratne[4], Robert Vautard[5], Dim Coumou[1,6], Kristie L. Ebi[7], Julie Arrighi[8,9,10], Roop Singh[8], Maarten van Aalst[8,9,11], Carolina Pereira Marghidan[9], Michael Wehner[12], Wenchang Yang[13], Sihan Li[14], Dominik L. Schumacher[4], Mathias Hauser[4], Rémy Bonnet[5], Linh N. Luu[1], Flavio Lehner[15,16], Nathan Gillett[17], Jordis S. Tradowsky[18,19], Gabriel A. Vecchi[13,20], Chris Rodell[21], Roland B. Stull[21], Rosie Howard[21], and Friederike E. L. Otto[14]

[1]Royal Netherlands Meteorological Institute (KNMI), De Bilt, The Netherlands
[*]These authors contributed equally to this work.
[2]Atmospheric, Oceanic and Planetary Physics, University of Oxford, UK
[†]deceased, 12 October 2021
[3]Pacific Climate Impacts Consortium, University of Victoria, Victoria, V8R4J1, Canada
[4]Institute for Atmospheric and Climate Science, Department of Environmental Systems Science, ETH Zurich, Zurich, Switzerland
[5]Institut Pierre-Simon Laplace, CNRS, Sorbonne Université, Paris, France
[6]Institute for Environmental Studies (IVM), VU Amsterdam, The Netherlands
[7]Center for Health and the Global Environment, University of Washington, Seattle WA, USA
[8]Red Cross Red Crescent Climate Centre, The Hague, the Netherlands
[9]Faculty of Geo-Information Science and Earth Observation (ITC), University of Twente, Enschede, the Netherlands
[10]Global Disaster Preparedness Center, American Red Cross, Washington DC, USA
[11]International Research Institute for Climate and Society, Columbia University, New York, USA
[12]Lawrence Berkeley National Laboratory, Berkeley, California USA
[13]Department of Geosciences, Princeton University, Princeton, 08544, USA
[14]School of Geography and the Environment, University of Oxford, UK
[15]Department of Earth and Atmospheric Sciences, Cornell University, USA
[16]Climate and Global Dynamics Laboratory, National Center for Atmospheric Research, USA
[17]Canadian Centre for Climate Modelling and Analysis, Environment and Climate Change Canada, Victoria, BC, Canada
[18]Deutscher Wetterdienst, Regionales Klimabüro Potsdam, Potsdam, Germany
[19]Bodeker Scientific, Alexandra, New Zealand
[20]The High Meadows Environmental Institute, Princeton University, Princeton, 08544, USA
[21]Department of Earth, Ocean, and Atmospheric Sciences, The University of British Columbia, Vancouver, Canada

**Correspondence:** Sjoukje Philip (sjoukje.philip@knmi.nl); Sarah Kew (sarah.kew@knmi.nl)

**Abstract.** Towards the end of June 2021, temperature records were broken by several degrees Celsius in several cities in the Pacific northwest areas of the U.S. and Canada, leading to spikes in sudden deaths, and sharp increases in emergency calls and hospital visits for heat-related illnesses. Here we present a multi-model, multi-method attribution analysis to investigate to what extent human-induced climate change has influenced the probability and intensity of extreme heatwaves in this region.

Based on observations, modelling and a classical statistical approach, the occurrence of a heatwave defined as the maximum daily temperatures (TXx) observed in the area 45 °N–52 °N, 119 °W–123 °W, was found to be virtually impossible without

human-caused climate change. The observed temperatures were so extreme that they lay far outside the range of historical temperature observations. This makes it hard to state with confidence how rare the event was. Using a statistical analysis that assumes that the heatwave is part of the same distribution as previous heatwaves in this region, led to a first order estimation of
the event frequency of the order of once in 1000 years under current climate conditions. Using this assumption and combining the results from the analysis of climate models and weather observations, we found that such a heatwave event would be at least 150 times less common without human-induced climate change. Also, this heatwave was about 2°C hotter than a 1 in 1000-year heatwave would have been in 1850–1900, when global mean temperatures were 1.2°C cooler than today. Looking into the future, in a world with 2°C of global warming (0.8°C warmer than today), a 1000-year event would be another degree
hotter. Our results provide a strong warning: our rapidly warming climate is bringing us into uncharted territory with significant consequences for health, well-being, and livelihoods. Adaptation and mitigation are urgently needed to prepare societies for a very different future.

## 1  Introduction

During the last days of June 2021, Pacific northwest areas of the U.S. and Canada experienced temperatures never previously
observed, with temperature records broken in multiple cities by several degrees Celsius. Temperatures far above 40 °C (104 °F) occurred on Sunday 27 to Tuesday 29 June (Figs 1a,b for Monday) in the Pacific northwest areas of the U.S. and western Provinces of Canada, with the maximum warmth moving from the western to the eastern part of the domain from Monday to Tuesday. The anomalies relative to the daily maximum temperature climatology for the time of year reached 16°C to 20 °C (Figs 1c,d). It is noteworthy that these record temperatures occurred one whole month before the climatologically warmest part
of the year (end of July, early August), making them particularly exceptional. Even compared to the average annual maximum temperatures in other years independent of the month considered, the recent event exceeds those temperatures by about 5 °C (Figure 2). Records were shattered in a very large area, including the village of Lytton where a new all-time Canadian temperature record of 49.6 °C was set on June 29, and where wildfires spread on the following day.

Given that the observed temperatures were so far outside historical experiences and occurred in a region with only about
50% household air conditioning penetration, large impacts on health were expected. In British Columbia, Canada, there were an estimated 619 heat-related excess deaths putting the event among the deadliest weather related events in Canada [1]. There were an estimated 196 extra deaths in Washington and 100 in Oregon. [2] [3]

Below we investigate the role of human-induced climate change in contributing to the likelihood and intensity of this extreme heatwave, following an established approach to multi-model multi-method extreme event attribution (Philip et al., 2020; van
Oldenborgh et al., 2021). We focus the analysis on the maximum temperatures in the region where most people were affected by the heat (45 °N–52 °N, 119 °W–123 °W) including the cities of Seattle, Portland, and Vancouver. While the extreme

---

[1] https://www2.gov.bc.ca/assets/gov/birth-adoption-death-marriage-and-divorce/deaths/coroners-service/death-review-panel/extreme_heat_death_review_panel_report.pdf

[2] https://www.opb.org/article/2021/08/06/oregon-june-heat-wave-deaths-names-revealed-medical-examiner/

[3] https://doh.wa.gov/emergencies/be-prepared-be-safe/severe-weather-and-natural-disasters/hot-weather-safety/heat-wave-2021

## Maximum temperature

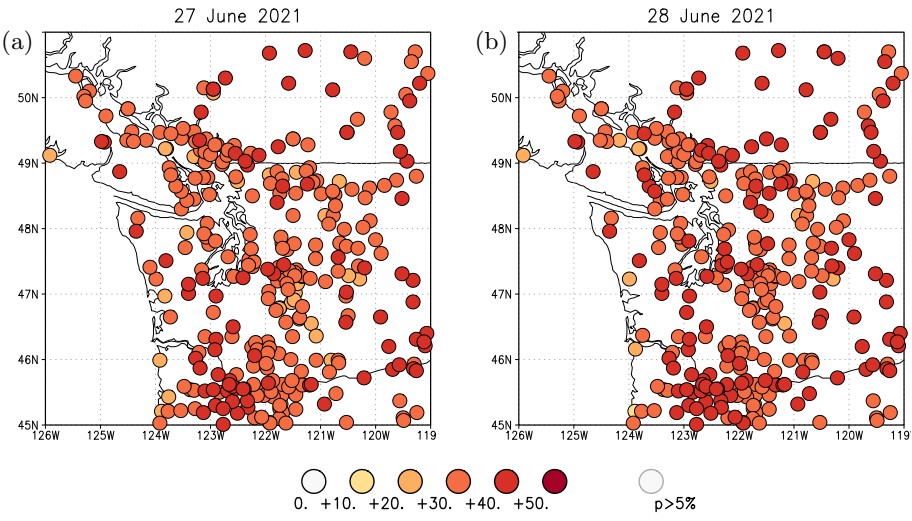

## Maximum temperature anomaly

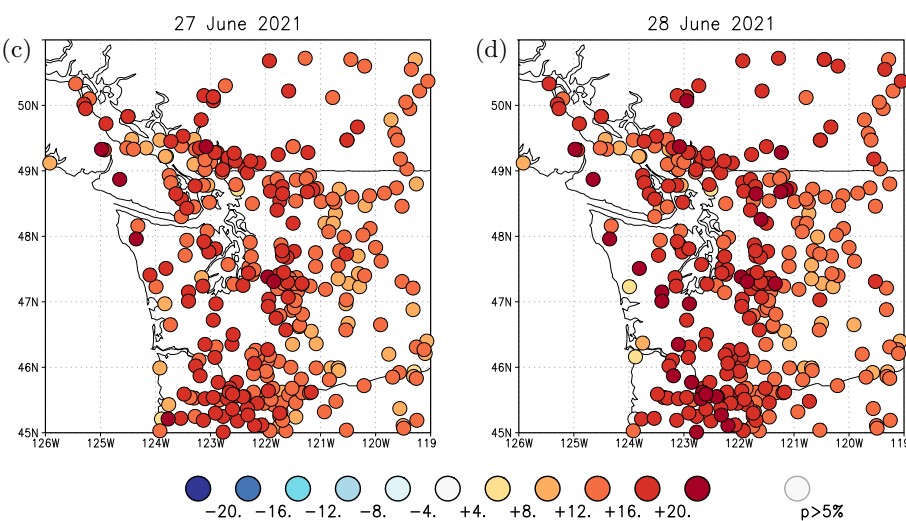

**Figure 1.** a) observed temperatures on 27 June 2021, b) 28 June 2021, c,d) same for anomalies relative to the climatological mean of the whole station records for these specific days of the year. Source: GHCN-D

heat was an important driver of the observed impacts, it is important to note that these impacts strongly depend on exposure, vulnerability, and other climatological variables beyond temperature. In addition to the attribution of the extreme temperatures, we qualitatively assess whether meteorological drivers and antecedent conditions played an important role in the observed
extreme temperatures in Section 6.

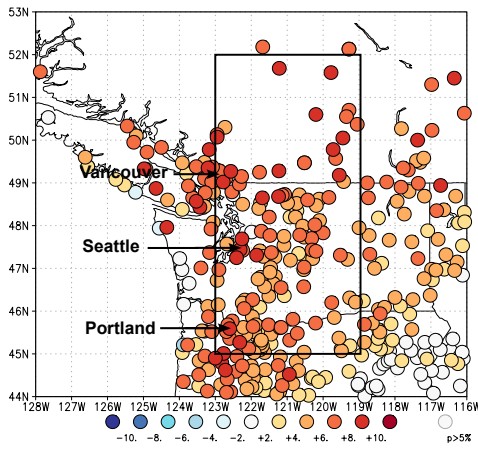

**Figure 2.** Anomalies of 2021 highest daily maximum temperature (TXx) relative to the mean TXx of the whole time series of each station. The black box indicates the study region. Source: GHCN-D

## 1.1 Event definition

For the attribution analysis, a definition of the event is required. As there are several ways to define an event, this step requires some expert judgement. Within this study, we will analyse the daily maximum temperatures, which characterised the event and dominated headlines in the large number of media reports describing the heatwave and its associated impacts. We therefore
define the event based on the annual maximum of daily maximum temperature, TXx. Here, we first average over the region and then take the annual maximum. Other options for variables that could be selected for analysis include e.g. 3-day averaged temperatures, as there is some evidence that longer time scales better describe the health impacts (e.g., D'Ippoliti et al., 2010) or high minimum temperatures which also have strong impacts on human health. However, TXx is a standard heat impact index and thus the results can easily be compared to other studies. We intentionally focus on one event definition to keep this
analysis succinct and its results easy to communicate, choosing TXx, which not only characterises the extreme character of the event but is also readily available in climate models allowing us to use a large range of different models. Recognising that the WMO standard definition of a heatwave is a multi-day measure associated with persistent heat, we additionally considered the annual maxima of 5-day and 10-day averages of daily maximum temperatures, TX5x and TX10x, for observations only. Trends in these time series, as seen by the intensity change, turn out to be very similar to the results for TXx presented in Sect. 3.
For the spatial scale of the event we consider the area 45°N–52°N, 119°W–123°W. This covers the more populated region around Portland, Seattle and Vancouver that were impacted heavily by the heat (with a total population of over 9.4 million in their combined metropolitan areas), but excludes the rainforest to the west and arid areas to the east. Note that this spatial event definition is based on the expected and reported human impacts rather than on the meteorological extremity. Besides the analysis for this region, we also analysed long and homogeneous observational records of three stations in Portland, Seattle
and Vancouver.

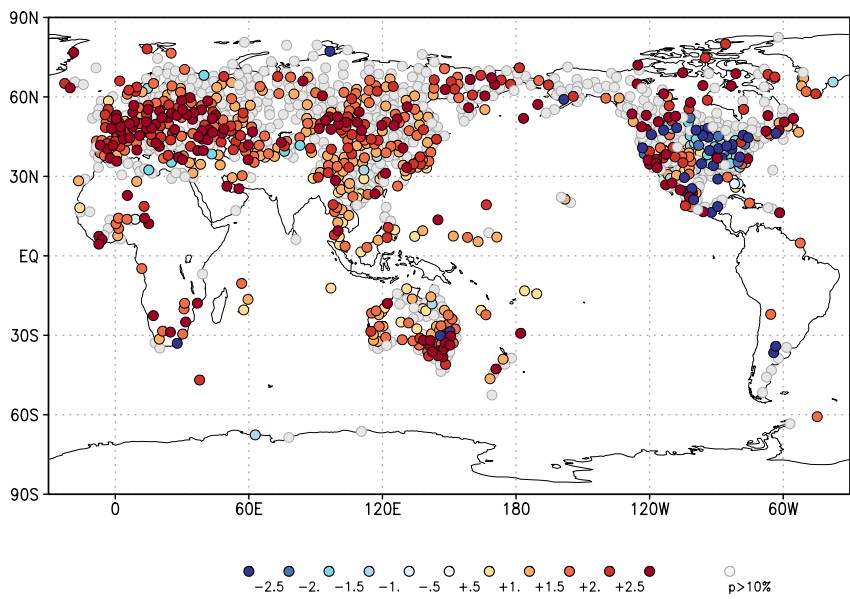

**Figure 3.** Trends in the highest daily maximum temperature of the year in the GHCN-D station data. Stations are selected to have at least 50 years of data and to be at least 2°apart. The local trends are defined by their regression on the global mean temperature, and shown in units of multiples of the global mean temperature rise. Source: GHCN-D

## 1.2 Previous trends in heatwaves

Figure 3 shows the observed trends in TXx in the GHCN-D dataset over 1900–2019. The stations were selected on the basis of a long time series of at least 50 years of data, and were required to be at least 2°apart. The trend is defined as the regression on the global mean temperature, so the numbers represent how much slower or faster the temperature has changed compared to the global mean temperature. Individual stations with different trends than nearby stations usually have inhomogeneities in the observational method or local environment. There are large positive trends in heatwaves in Europe. These are not yet fully understood or adequately represented in climate models (Vautard et al., 2020). The negative trends in eastern North America and parts of California are well-understood to be the result of land use changes, irrigation and changes in agricultural practice (Donat et al., 2016, 2017; Thiery et al., 2017; Cowan et al., 2020). The Pacific northwest shows positive trends twice as large as the global mean temperature trend up to 2019.

## 2 Data and methods

### 2.1 Observational data

The dataset used to represent the heatwave is the ERA5 reanalysis (Hersbach et al., 2019) from the European Centre for Medium-range Weather Forecasts (ECMWF) at 0.25°resolution. A very rapid analysis performed directly after the heatwave and published on https://www.worldweatherattribution.org/western-north-american-extreme-heat-virtually-impossible-without-human-caused-climate-change/ used ERA5 extended by the ECMWF operational analysis and the ECMWF forecast. The differences in the heatwave amplitude between the rapid analysis and the updated analysis presented here are minor and do not affect the rounded estimate of the return period used in this study. Therefore, the analysis results do not require updating.

Temperature observations were used to assess probability ratios and return periods associated with the event for three major cities in the study area: Portland, Seattle, and Vancouver. Observing sites were chosen based on (i) the availability of long homogenised historical records, (ii) their ability to represent the severity of the event by avoiding sites within the proximity of large water bodies, and (iii) their representativeness for populous areas of each city to better illuminate impact on inhabitants.

For Portland, the Portland International Airport National Weather Service station was used, which has continuous observations over 1938–2021. The airport is located close to the city centre, adjacent to the Columbia River. The river's influence is thought to be small. For Seattle, Seattle-Tacoma International Airport was chosen, which has almost continuous observations between 1948 and 2021, making it one of the stations with the longest records in the Seattle area. This location is further inland and thus avoids the influence of Lake Washington that affects downtown Seattle. Two long records exist adjacent to downtown Vancouver, but they are both very exposed to the Georgia Strait that influenced observations due to local onshore flow during the peak of the event. Thus the time series from a site further inland at New Westminster was selected, which starts in 1875 but contains data gaps in 1882–1893, 1928, and 1980–1993.

The data for Portland International Airport and Seattle-Tacoma International Airport were gathered from the Global Historical Climatology Network Daily (GHCND; Menne et al., 2012) while data for New Westminster were gathered from the Adjusted Homogenized Canadian Climate Dataset (AHCCD) for daily temperature (Vincent et al., 2020). This station's record is a composite of data from three locations in two nearby cities as location changes took place in 1966 and 1980. From 1874 to 1966, the station operated at an elevation of 118 m above sea level near the centre of New Westminster. In 1966, the station was moved about 2 km east and to an elevation of 18 m. The portion of the homogenised record from 1980 onward is from Pitt Meadows, BC, located about 14 km east of the previous location at an elevation of 5 m. Using a composite station is non-ideal given the potential influence of local micro-climatic effects and particularly the increasing distance from the Strait of Georgia which exhibits a cooling effect to sites in its vicinity. Use of data from this composite site may increase the uncertainty of our analysis, but given the magnitude of the signal and the consistency of results among the datasets presented here (and analysis of other temperature datasets from BC, not shown here), we do not expect this to substantially affect our results. The AHCCD dataset is updated annually and ends in 2020. Data for 2021 were appended from unhomogenized recent records from Environment and Climate Change Canada. Overlapping data for 2020 were compared between the two sources and found to be

identical except for several duplicate/missing observations. Such duplicate or missing data would not cause error in the present analysis because the records are complete for June 2021.

As a measure of anthropogenic climate change we use the global mean surface temperature (GMST), where GMST is taken from the National Aeronautics and Space Administration (NASA) Goddard Institute for Space Science (GISS) surface temperature analysis (Hansen et al., 2010; Lenssen et al., 2019, GISTEMP,). We apply a 4-yr running mean low-pass filter to

suppress the influence of ENSO and winter variability at high northern latitudes as these are unforced variations.

## 2.2    Model and experiment descriptions

In this study a variety of climate model simulations are analysed and will be described in the following paragraphs.

Model simulations from the 6th Coupled Model Intercomparison Project (CMIP6; Eyring et al., 2016) are assessed after combining the historical simulations (1850 to 2014) with the Shared Socioeconomic Pathway (SSP) projections (O'Neill et al.,

2016) for the years 2015 to 2100. Here, we only use data from SSP5-8.5, noting that SSPs are very similar over the period 2015–2021. Models are excluded if they do not provide the relevant variables, do not run from 1850 to 2100, or if they either include duplicate time steps or miss time steps. All available ensemble members are used. A total of 18 models (88 ensemble members), which fulfil these criteria and passed the validation tests (Section 4), are used.

In addition an ensemble of extended historical simulations from the IPSL-CM6A-LR model is used (see Boucher et al., 2020,

for a description of the model), which follows the CMIP6 protocol (Eyring et al., 2016). It is composed of 32 members, and the simulations cover the historical period (1850–2014). It has been extended until 2029 using all forcings from the SSP2-4.5 scenario, except for the ozone concentration which has been kept constant at its 2014 climatology, as it was not available at the time the extensions were generated. This ensemble is used to explore the influence of internal variability.

Furthermore we use the GFDL-CM2.5/FLOR model (Vecchi et al., 2014), which is a fully coupled climate model developed

at the Geophysical Fluid Dynamics Laboratory (GFDL). While the ocean and ice components have a horizontal resolution of only $1°$, the resolution of atmosphere and land components is about 50 km and might therefore provide a better simulation than coarser model simulations of certain extreme weather events (Baldwin et al., 2019). The data used in this study cover the period from 1860 to 2100, and combine historical and RCP4.5 experiments driven by transient radiative forcings from CMIP5 (Taylor et al., 2011).

We also examine five ensemble members of the AMIP experiment (1871–2019) from the GFDL-AM2.5C360 (Yang et al., 2021; Chan et al., 2021), which consists of the atmosphere and land components of the FLOR model but with finer horizontal resolution of 25 km for a potentially better representation of extreme events.

Further, we use simulations of the Climate of the 20th Century Plus Project (C20C+) project, which was designed specifically for event attribution studies (Stone et al., 2019). C20C+ simulations use models of the atmosphere and land with prescribed

sea surface temperatures and sea ice concentrations, similar to the design of AMIP experiments. To quantify the impact, if any, on extreme events, participating models were run following the AMIP protocol, and additional sets of counterfactual simulations were performed with anthropogenic influence removed. The distribution of TXx in the study area was examined for three C20C+ models, i.e. CAM5.1, MIROC5 and HadGEM3-A-N216 and compared to that of the ERA5 reanalysis. Only

the Community Atmospheric Model (CAM5.1, Neale et al. (2010)), run at ~1°resolution, satisfied the requirements of this study in the statistical description of heat extremes. The actual world ensemble consists of 99 simulations of mixed duration all ending in 2018, resulting in a sample size of 4090 years. A counterfactual world ensemble of similar size consists of 89 simulations resulting in a sample size of 3823 years.

## 2.3  Statistical methods

A more detailed description of the statistical methods is given in (Philip et al., 2020; van Oldenborgh et al., 2021). Here we give a description of the most important aspects.

As discussed in Section 1.1, we analyse the annual maximum of daily maximum temperatures (TXx) averaged over 45°N-52°N, 119°W-123°W. Initially, we analyse reanalysis data and station data from sites with long data records. Next, we analyse climate model output of TXx. We follow the steps corresponding to the World Weather Attribution (WWA) protocol for event attribution which for the analysis consist of: (i) trend calculation from observations; (ii) model validation; (iii) multi-method multi-model attribution and (iv) synthesis of the attribution statement. Steps (i) and (iii) are briefly outlined below, step (ii) is explained in Section 4 and we elaborate on step (iv) in Section 5

For the event under investigation we calculate the return periods, probability ratio (PR) and change in intensity ($\Delta$T) as a function of GMST, where PR is defined as $\mathrm{PR} = p_1/p_0$, with $p_1$ the probability of an event as strong as or stronger than the extreme event in the current climate and $p_0$ the probability of such an event in a counterfactual climate without anthropogenic emissions. The two climates we compared are defined by the GMST of the event year 2021 and a GMST value representative of the climate of the late nineteenth century, $-1.2$ °C relative to 2021 (1850–1900, based on the Global Warming Index https://www.globalwarmingindex.org).

To statistically model the selected event, we use a GEV distribution (see e.g., Coles, 2001) that shifts with GMST, i.e., the location parameter has a term proportional to GMST and the scale and shape parameters are assumed constant. Uncertainties corresponding to the statistical-model uncertainty, are obtained using a non-parametric bootstrap procedure. With this GEV distribution, first the PR and intensity change are calculated from observations, as well as the return period in the current climate. Next, the return period is used as a threshold to specify the event magnitude for the models. For this return period, the PRs and intensity changes between 2021 and the counterfactual climate are calculated from different models. This is, however, only done for models that pass our validation tests on the seasonal cycle, the spatial pattern of the climatology, and the scale and shape parameters of the GEV distribution, see Section 4. Finally, both observational and model results are synthesised into a consistent attribution statement, see Section 5. For models (except IPSL-CM6A-LR and CAM5.1), we additionally analyse the PR between the current climate and a future climate at +2°C above the 1850–1900 reference, which is equivalent to +0.8°C above the current climate of 2021. For this analysis of future change we use model data up to about 2050 or when the model GMST reaches +0.8 °C compared to now.

The CMIP6 data are analysed using the same statistical models as the main method. However, for practical reasons, the parameter uncertainty is estimated in a Bayesian setting using a Markov Chain Monte Carlo (MCMC) sampler instead of a bootstrapping approach (see Ciavarella et al., 2021, for another example of its application).

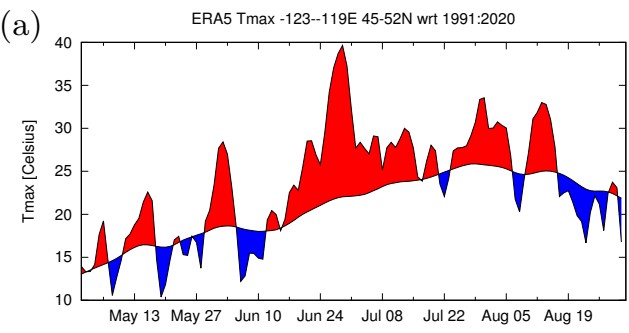

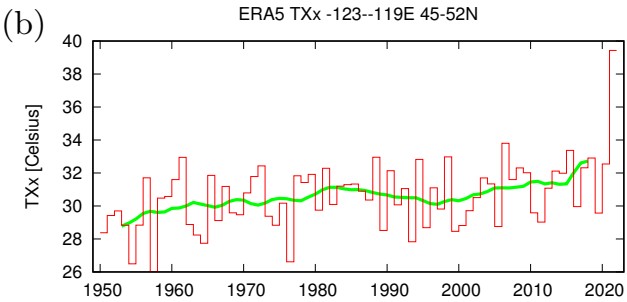

**Figure 4.** a) Time series for May-Aug 2021 of the maximum daily temperature averaged over the study area based on ERA5, with positive and negative departures from the 1991–2020 climatological mean of daily maximum temperature shaded red and blue, respectively. b) Annual maximum of the index series with a 10-yr running mean (green line). Source: ERA5.

## 3  Observational analysis: return period and trend

Time series of various aspects of the main index are shown in Figure 4: a) the daily maximum temperature (Tmax) evolution from ERA5 (from 1 May to 31 Aug); and b) annual maximum of Tmax — the index series. The value for 2021, 39.7 °C, is 5.7 °C above the previous record of 34.0 °C. This extremely large increase leads to difficulties in the statistical analysis described below. There are two possible sources of this extreme jump in peak temperatures. The first is that an event with very low probability occurred — the statistical equivalent of "really bad luck". An event with very low probability could also have occurred in the pre-industrial climate but its amplitude would have been increased by climate change in the current climate which already includes about 1.2°C of global warming. The second option is that strong nonlinear interactions and feedbacks took place in this event, amplifying the intensity of the extreme, This would mean that climate change is acting to exacerbate extreme heatwaves more rapidly than implied by a GEV with a location parameter which scales linearly with GMST. In this case, the event would not belong to the "same population" as the other ones and we would not expect the method applied here to be successful. This second possibility requires further investigation. While we keep this possibility in mind, we assume the first option within this study.

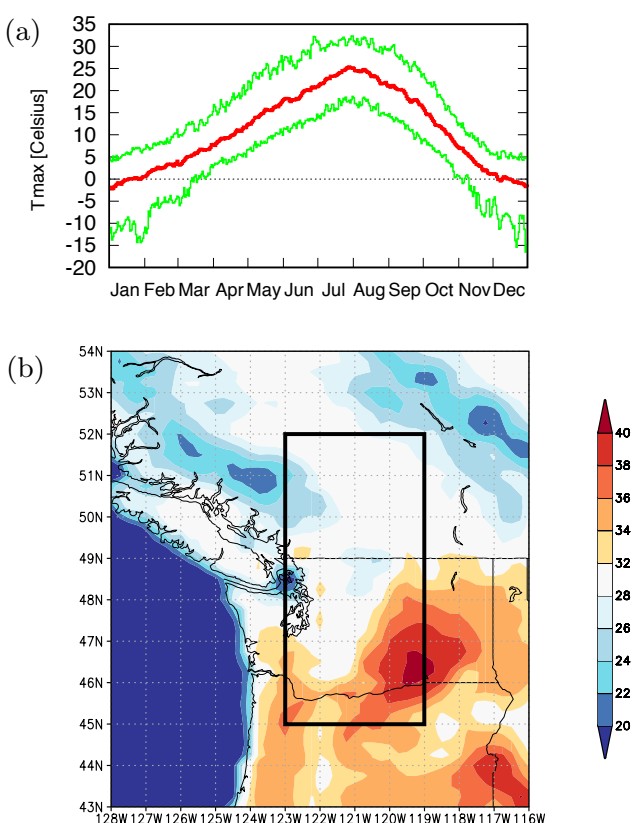

**Figure 5.** a) Seasonal cycle of Tmax averaged over the land points of 45 °N–52 °N, 119 °N–123 °N, showing the 1950–2021 mean (red) and 2.5% and 97.5% percentiles of the distribution (green). b) Spatial pattern of the 1950–2021 mean of the annual maximum of Tmax (multi-year mean TXx) at each grid point. Source: ERA5.

In Figure 5a we show the seasonal cycle of the daily maximum temperature averaged over the index region and in Figure 5b we show the spatial pattern of TXx averaged over many years at each grid point individually. These two metrics are used in the model validation procedure, see Section 4.

## 3.1    Analysis of station and gridded data

Figure 6 shows the analysis of the gridded (ERA5) data using the ordinary extreme value analysis applied for attribution studies within WWA. This approach excludes data of the extreme event of interest from the statistical fit to avoid selection bias, especially when the study area has been defined based on the extent of the extreme event. Figure 6a shows the observed TXx as a function of smoothed GMST and shows that the value observed in 2021 is far outside the range of any values observed to date. The distribution of TXx including data up to 2020 is described very well by a GEV distribution that has linearly warmed at a rate about twice as fast as the GMST, see Figure 6b. The warming rate is consistent with expectations, as summer temperatures over continents increase faster than the global mean. The GEV fit has a negative shape parameter $\xi$,

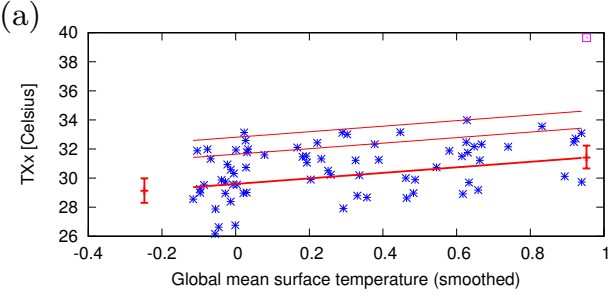

(a)

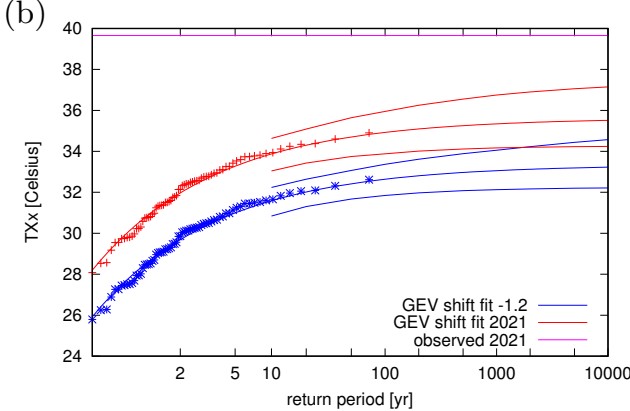

(b)

**Figure 6.** GEV fit with constant scale and shape parameters, and location parameter shifting proportional to GMST of the index series. No information from 2021 is included in the fit. (a): the observed TXx as a function of the smoothed GMST. The thick red line denotes the location parameter, the thin red lines the 6-yr and 40-yr return period levels. The June 2021 observation is highlighted with the magenta square and is not included in this fit. (b): Return period plots for the climate of 2021 (red) and a climate with GMST 1.2 °C cooler (blue). The red and blue lines indicate the best fit and the 95% confidence intervals, the magenta line shows the observed value. The past observations are shown twice: once shifted up to the current climate and once shifted down to the climate of the late nineteenth century. Source: ERA5, fit: KNMI Climate Explorer.

which implies a finite tail, and hence an upper bound, here at about 35.5±1.3 °C ($2\sigma$ uncertainty). However, the value observed in 2021, 39.7 °C, is far above this upper bound. Therefore, this GEV fit with constant shape and scale parameters that excludes all information about 2021 does not provide a valid description of heatwaves in the area.

An alternative to the standard approach for which no information of the event under study is used (to avoid a selection bias), is to use the information that it actually happened, yet without including the value observed in 2021 in the fit. Specifically, we again assume that the data up to 2020 can be described by a GEV distribution with constant scale and shape parameters, but we reject all GEV models in which the upper bound is below the value observed in 2021. In other words, we enforce that the fit parameters are within a subset of parameters that are compatible with the 2021 event. The result is shown in Figure 7. While the distribution now includes the 2021 event, the fit to the data up to and including the year 2020 is noticeably worse

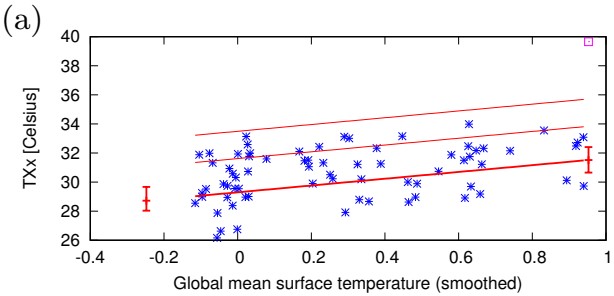

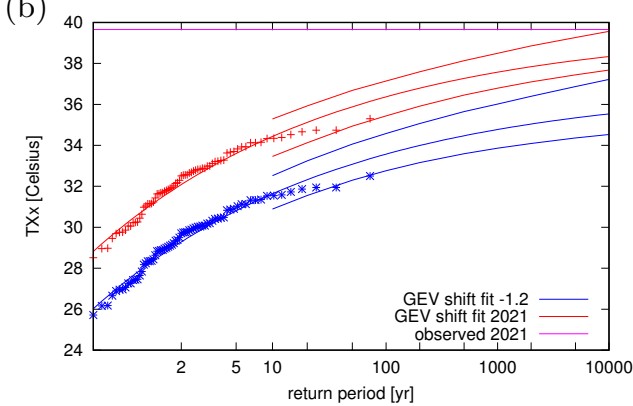

**Figure 7.** As Figure 6 but demanding the 2021 event is possible in the fitted GEV function, i.e., the upper bound is higher than the value observed in 2021.

than when not taking 2021 into account. The return period for the 2021 event under these assumptions still has a lower bound of 10,000 years in the current climate. The fit differs from the previous one mainly in the shape parameter, which is now much less negative (about $-0.2$ instead of $-0.4$). This shifts the upper bound to higher values. The fit also gives a somewhat higher trend parameter.

The third possibility is to fit the GEV distribution over all available data, including observations from 2021. This approach implicitly assumes that the 2021 event is drawn from the same population. We typically do not make this assumption in cases where we intentionally select a specific region in order to maximise the extremity (i.e. return period) of the event to avoid a selection bias. Note however that we may have overestimated the return period by excluding the event, a question left for future investigations. In intentionally choosing the region with the largest (rarest) return period for the 2021 event, the extreme value

is drawn from a different distribution and cannot be included. However, this is only partly the case here. We did choose the general region because the temperatures were exceptional there, however, we based the choice of subregion for the analysis on population density and type of terrain —parameters that are independent of the heatwave. The benefit of this approach is that it uses all available information. With this approach we still assume this was an event happening by chance, that is, the behaviour is in line with that of a chaotic deterministic system and by chance we observe a low-probability event in this short time series.

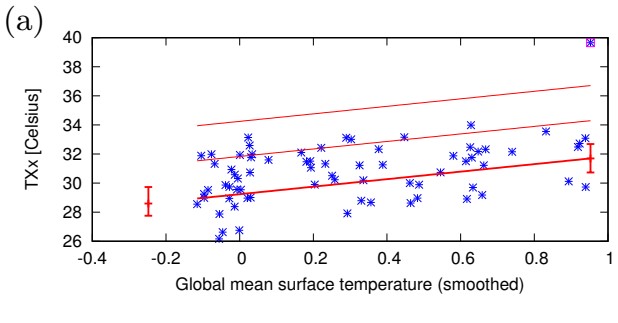

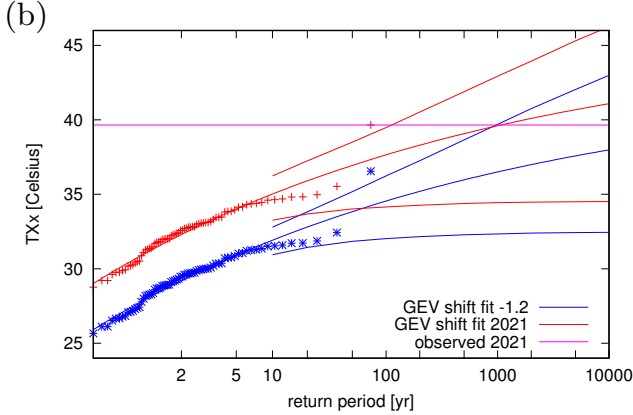

**Figure 8.** As for Figure 6 but including data from the 2021 heatwave into the fit.

Given the extremity of the event and the relatively low number of data, a robust GEV fit is hard to obtain, and the appropriateness of the method is difficult to assess. However, the application of this classical method in this case is interesting provided we keep in mind the assumptions we make. While we acknowledge that none of the three possibilities to fit a GEV distribution is fully satisfying, we decided on using this third approach to estimate the return period leading to an estimate of 1,000 years (95% CI >100 yr). Follow-up research will be necessary to investigate the potential reasons for this exceptional event and the consequences for assumptions for these fits (see also the discussion in Sections 6 and 8). Also, further research is needed into the limitations of standard GEV analysis on annual maxima with short records and very extreme values. Climate model large ensembles offer a future test bed to investigate the robustness of the method in light of current limitations.

The fit including observations from 2021 gives a 95% CI of 1.4 K to 1.9 K for the scale parameter $\sigma$ and $-0.5$ to 0.0 for the shape parameter $\xi$. These values are used for the model validation in Section 4.

The observational analysis results, i.e., the comparison of the fit for 2021 and for a pre-industrial climate, show an increase in intensity of TXx of $\Delta T = 3.1\ °C$ (95% CI: 1.2 to 4.8 °C) and a probability ratio PR of 390 (3.2 to $\infty$).

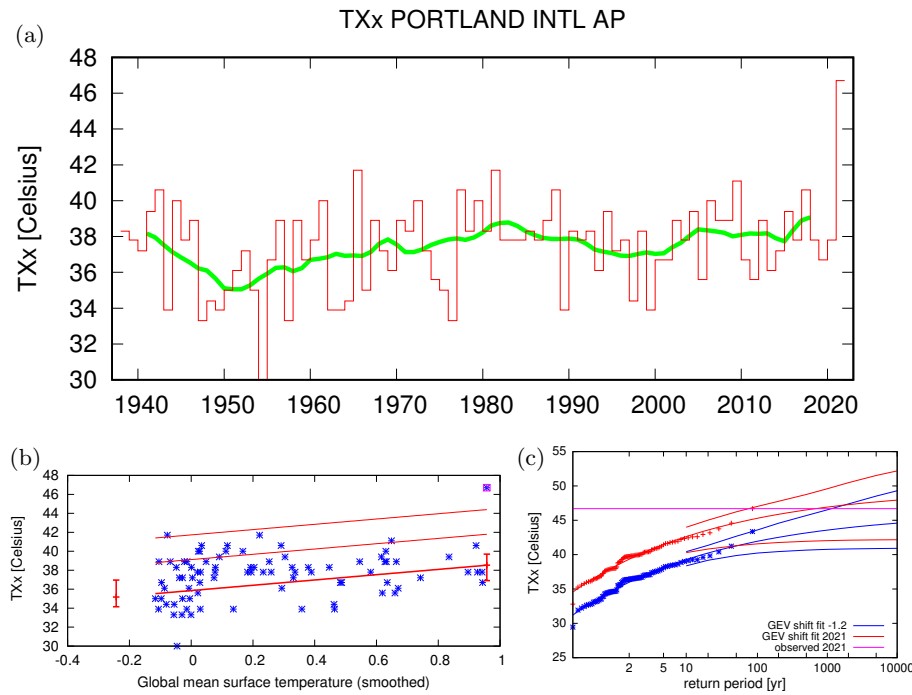

**Figure 9.** (a) time series of observed highest daily maximum temperature of the year at Portland International Airport. (b,c): as Figure 8 but for the station data at Portland International Airport. Source: data GHCN-D, fit: KNMI Climate Explorer.

### 3.2 Analysis of temperature in Portland, Seattle and Vancouver

To represent Portland we chose the International Airport station, which is located on the northern edge of the city and has been collecting data since April 1938; the data are in the GHCN-D v2 database. Figure 9 (top panel) shows the annual maxima of the Portland station time series The record before 2021 was 41.7 °C in 1965 and 1981, and TXx reached 46.7 °C in 2021, so the previous record was broken by 5.0 °C.

We fit a GEV distribution to this data, including 2021 (Figure 9, lower panels). It gives a return period of 700 yr for the 2021 event with a lower bound of 70 yr. For the PR we can only give a lower bound (6), since the best estimate is infinite. This corresponds to an increase in TXx of 3.4 °C with a large uncertainty of 0.3 to 5.3 °C. The large uncertainties are due to the somewhat shorter time series and large variability at this station.

In Seattle, the only station with a sufficiently long time series that includes 2021 is Seattle-Tacoma International Airport. It is located ∼15 km south of the city but has similar terrain, without the proximity to water of the city itself. The previous record was 39.4 °C in 2009, and in 2021 it reached 42.2 °C. This is still a large increase of 2.8 °C over the previous record. The event was also not quite as improbable, with a return period of 300 yr (lower bound 40 yr) in the current climate (Figure 10). The PR is again infinite with a lower bound of 7, and the increase in temperature from a late nineteenth century climate is 3.8 °C (0.7 °C to 5.7 °C).

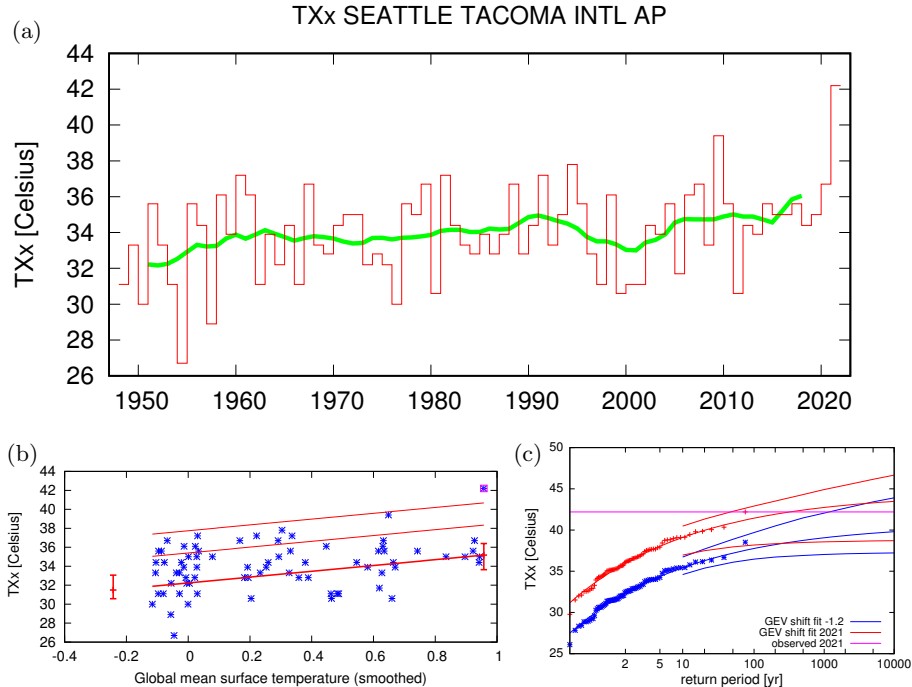

**Figure 10.** as Figure 9 but for the station data at Seattle-Tacoma International Airport.

In the Vancouver area, the most representative station with the least missing data is New Westminster. It has data from 1875 to 2021 with a few gaps. The previous record was 37.6 °C in 2009, and in 2021 a temperature of 41.4 °C was observed — 4.0 °C warmer. A GEV fit including 2021 gives a return period of 1000 yr with a lower bound of 70 yr (Figure 11). The PR is infinite with a lower bound of 170, and the temperature increased by 3.4 °C (1.9 °C to 5.5 °C).

## 4 Model evaluation and analysis

In this section we show the results of the model validation. The validation criteria assess the similarity between the modelled and observed seasonal cycle, the spatial pattern of the climatology, and the scale and shape parameters of the GEV distribution. The assessment results in a label "good", "reasonable" or "bad", according to the criteria defined in Ciavarella et al. (2021). In this study, we use models that are labelled "good" or "reasonable". However, if five or more models are classified as "good" within a particular model set such as the CMIP6 models, then we do not include all of the "reasonable" models but only those that pass the specific test on fit parameters as "good". Table 1 shows the model validation results. The full table including the models that did not pass the validation tests is given in Table 3. In total 21 models and a combined 224 ensemble members passed the validation tests.

Next, we show probability ratios and change in intensity ΔT for models that pass the validation tests and we also include the threshold values for a 1-in-1000 year event (Table 2). Results are given both for changes in the current climate (1.2°C)

**Table 1.** Validation results for models that passed the validation tests on seasonal cycle, spatial pattern and fitted GEV scale parameter and shape parameter (sigma). Observations in italic.

| Model / *Observations* (number of members) | Seasonal cycle | Spatial pattern | Sigma | Shape parameter | Conclusion |
|---|---|---|---|---|---|
| *ERA5* | | | *1.70 (1.40 ... 1.90)* | *-0.200 (-0.500 ... 0.00)* | |
| GFDL-CM2.5/FLOR historical-rcp45 (5) | good | good | 2.01 (1.84 ... 2.17) | -0.201 (-0.272 ... -0.144) | reasonable, include as different experiment than most other models |
| ACCESS-CM2 historical-ssp585 (2) | good | good | 1.86 (1.71 ... 2.02) | -0.200 (-0.260 ... -0.120) | good |
| AWI-CM-1-1-MR historical-ssp585 (1) | good | good | 1.50 (1.35 ... 1.69) | -0.200 (-0.280 ... -0.110) | good |
| CNRM-CM6-1 historical-ssp585 (1) | good | good | 1.54 (1.39 ... 1.72) | -0.210 (-0.290 ... -0.100) | good |
| CNRM-CM6-1-HR historical-ssp585 (1) | good | good | 1.48 (1.33 ... 1.66) | -0.190 (-0.270 ... -0.100) | good |
| CNRM-ESM2-1 historical-ssp585 (1) | good | good | 1.71 (1.54 ... 1.92) | -0.180 (-0.250 ... -0.0900) | good |
| CanESM5 historical-ssp585 (50) | good | reasonable | 1.79 (1.76 ... 1.82) | -0.180 (-0.190 ... -0.170) | reasonable, include because statistical parameters good |
| EC-Earth3 historical-ssp585 (3) | good | good | 1.87 (1.76 ... 2.00) | -0.220 (-0.270 ... -0.170) | good |
| FGOALS-g3 historical-ssp585 (3) | good | reasonable | 1.80 (1.69 ... 1.92) | -0.180 (-0.210 ... -0.140) | reasonable, include because statistical parameters good |
| GFDL-CM4 historical-ssp585 (1) | good | good | 1.43 (1.29 ... 1.62) | -0.210 (-0.300 ... -0.110) | good |
| INM-CM4-8 historical-ssp585 (1) | good | good | 1.63 (1.46 ... 1.83) | -0.210 (-0.300 ... -0.110) | good |
| INM-CM5-0 historical-ssp585 (1) | good | good | 1.80 (1.63 ... 2.03) | -0.240 (-0.310 ... -0.140) | good |
| IPSL-CM6A-LR historical-ssp585 (6) | good | reasonable | 1.79 (1.71 ... 1.88) | -0.220 (-0.250 ... -0.180) | reasonable, include because statistical parameters good |
| MIROC-ES2L historical-ssp585 (1) | reasonable, peaks early | reasonable | 1.46 (1.31 ... 1.65) | -0.190 (-0.300 ... -0.0900) | reasonable, include because statistical parameters good |
| MPI-ESM1-2-HR historical-ssp585 (2) | good | good | 1.49 (1.39 ... 1.62) | -0.250 (-0.310 ... -0.190) | good |
| MPI-ESM1-2-LR historical-ssp585 (10) | good | good | 1.63 (1.58 ... 1.69) | -0.260 (-0.280 ... -0.230) | good |
| MRI-ESM2-0 historical-ssp585 (2) | reasonable, peak too flat | good | 1.41 (1.30 ... 1.53) | -0.280 (-0.340 ... -0.220) | reasonable, include because statistical parameters good |
| NESM3 historical-ssp585 (1) | good | good | 1.48 (1.34 ... 1.67) | -0.290 (-0.370 ... -0.200) | good |
| NorESM2-MM historical-ssp585 (1) | good | good | 1.90 (1.70 ... 2.12) | -0.250 (-0.350 ... -0.140) | in between reasonable and good, include |
| IPSL-CM6A-LR historical-ssp245 (32) | good, from CMIP6 | reasonable, from CMIP6 | 1.69 (1.64 ... 1.75) | -0.220 (-0.250 ... -0.200) | reasonable, observed GMST used, include |
| CAM5-1-1degree C20C historical (99) | NA | NA | 1.70 (1.68 ... 1.72) | -0.176 (-0.172 ... -0.180) | good, values used with warming level 1.7 |

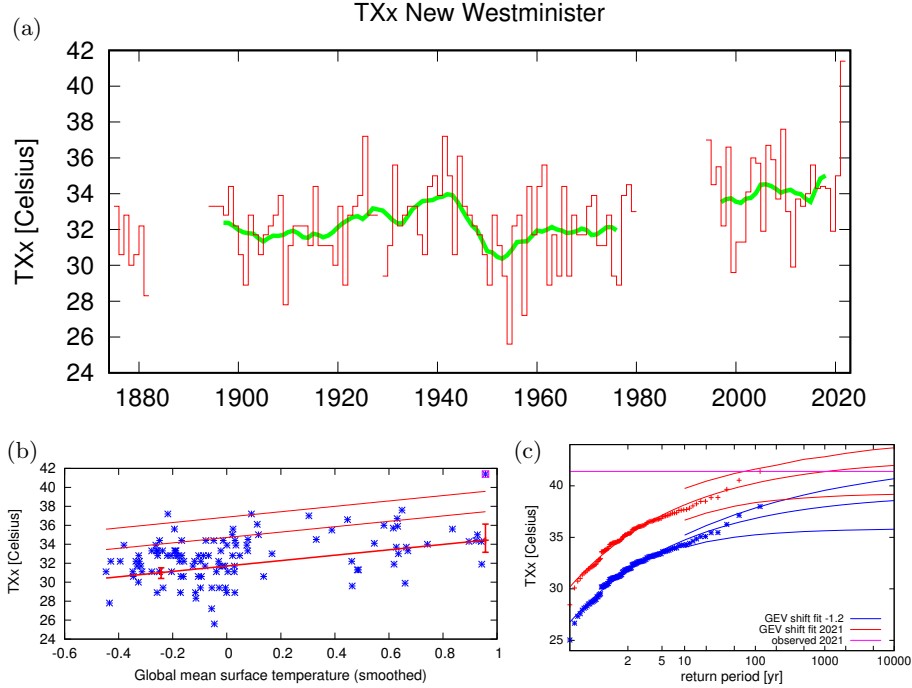

**Figure 11.** as Figure 9 but for the station data at New Westminster. Data source: AHCCD.

compared to the past (pre-industrial conditions) and, when available, for a climate at +2°C of global warming above pre-industrial climate compared with the current climate. The results are visualised in Section 5.

## 5 Hazard synthesis

In Sections 3 and 4 we calculated the probability ratio as well as the change in magnitude of the event in the observations and the models. In this section we combine these results to give an overarching synthesised attribution statement and present the results in Figures 12 and 13. The uncertainty due to differences in model set up and physics is represented by model spread — the average departure of each model from the mean model best estimate. This is added in quadrature to the model natural variability as white extensions to the light red bars in the synthesis figures. The uncertainty in the model average (bright red bar) consists of a weighted mean uncertainty, where the contribution from each model is inversely proportional to the uncertainty due to natural variability squared, plus the model spread term added in quadrature to the uncertainty in the weighted mean. Please see e.g. Kew et al. (2021) for more detailed information on the synthesis technique including how weighting is calculated for models. Observations and the model average are combined into a single result in two ways. Firstly, we compute the weighted average of models and observations: this is indicated by the magenta bar, see Figure 12. The weighting applied is the inverse square of the uncertainty (the width of the bright bars). Secondly, as there may be an additional model bias that is common to each model (and therefore cannot be detected from the spread of the models), we also show the more conservative estimate

**Table 2.** Analysis results showing the model threshold for a 1-in-1000 year event in the current climate, and the probability ratios and intensity changes for the present climate with respect to the past (labelled "past") and for the +2°C GMST future climate with respect to the present (labelled "future").

| Model / *Observations* (number of members) | Threshold | Probability ratio PR - past [-] | Change in intensity $\Delta T$ - past [°C] | Probability ratio PR - future [-] | Change in intensity $\Delta T$ - future [°C] |
|---|---|---|---|---|---|
| *ERA5* | *39.7 °C* | *3.5e+2 (3.2 ... ∞)* | *3.1 (1.1 ... 4.7)* | | |
| GFDL-CM2.5/FLOR historical-rcp45 (5) | 34 °C | 6.5e+2 (16 ... ∞) | 1.6 (1.2 ... 2.1) | 4.6 (3.4 ... 12) | 1.2 (1.0 ... 1.3) |
| ACCESS-CM2 historical-ssp585 (2) | 35 °C | 25 (2.3 ... ∞) | 1.1 (0.41 ... 1.9) | 45 (4.5 ... ∞) | 1.2 (0.96 ... 1.4) |
| AWI-CM-1-1-MR historical-ssp585 (1) | 36 °C | 1.1e+4 (6.6 ... ∞) | 1.6 (0.84 ... 2.3) | 2.8e+2 (5.5 ... ∞) | 1.3 (1.1 ... 1.6) |
| CNRM-CM6-1 historical-ssp585 (1) | 34 °C | 1.9 (0.0 ... ∞) | 0.22 (-0.51 ... 0.95) | 69 (3.4 ... ∞) | 1.1 (0.76 ... 1.3) |
| CNRM-CM6-1-HR historical-ssp585 (1) | 35 °C | 5.2e+2 (5.4 ... ∞) | 1.5 (0.73 ... 2.2) | 56 (4.1 ... ∞) | 1.3 (1.0 ... 1.5) |
| CNRM-ESM2-1 historical-ssp585 (1) | 38 °C | 1.5e+2 (3.6 ... ∞) | 1.6 (0.68 ... 2.6) | 15 (2.8 ... ∞) | 0.97 (0.64 ... 1.3) |
| CanESM5 historical-ssp585 (50) | 38 °C | 1.6e+3 (2.6e+2 ... 6.7e+4) | 2.0 (1.9 ... 2.1) | 62 (32 ... 1.5e+2) | 1.5 (1.4 ... 1.5) |
| EC-Earth3 historical-ssp585 (3) | 38 °C | 3.2e+2 (8.2 ... ∞) | 1.3 (0.88 ... 1.7) | 20 (5.2 ... 5.8e+2) | 1.2 (1.1 ... 1.4) |
| FGOALS-g3 historical-ssp585 (3) | 41 °C | 71 (8.5 ... 2.1e+8) | 1.5 (1.0 ... 2.0) | 17 (5.2 ... 2.2e+2) | 1.1 (0.87 ... 1.3) |
| GFDL-CM4 historical-ssp585 (1) | 31 °C | ∞ (14 ... ∞) | 2.1 (1.3 ... 3.0) | ∞ (16 ... ∞) | 1.7 (1.4 ... 1.9) |
| INM-CM4-8 historical-ssp585 (1) | 42 °C | ∞ (28 ... ∞) | 2.6 (1.7 ... 3.6) | 2.7e+3 (6.5 ... ∞) | 1.7 (1.4 ... 2.0) |
| INM-CM5-0 historical-ssp585 (1) | 41 °C | ∞ (14 ... ∞) | 2.2 (0.95 ... 3.3) | ∞ (12 ... ∞) | 1.6 (1.3 ... 2.0) |
| IPSL-CM6A-LR historical-ssp585 (6) | 34 °C | 1.5e+5 (50 ... ∞) | 1.7 (1.4 ... 2.0) | 2.4e+2 (16 ... ∞) | 1.3 (1.1 ... 1.4) |
| MIROC-ES2L historical-ssp585 (1) | 33 °C | 75 (1.3 ... ∞) | 1.2 (0.040 ... 2.3) | 12 (2.2 ... ∞) | 0.71 (0.41 ... 1.0) |
| MPI-ESM1-2-HR historical-ssp585 (2) | 34 °C | ∞ (27 ... ∞) | 1.4 (0.82 ... 1.9) | 4.8e+4 (12 ... ∞) | 1.2 (0.96 ... 1.4) |
| MPI-ESM1-2-LR historical-ssp585 (10) | 32 °C | ∞ (1.1e+11 ... ∞) | 1.6 (1.4 ... 1.9) | ∞ (1.8e+3 ... ∞) | 1.3 (1.2 ... 1.4) |
| MRI-ESM2-0 historical-ssp585 (2) | 32 °C | ∞ (1.3e+2 ... ∞) | 1.4 (0.86 ... 1.9) | 13 (4.9 ... 54) | 1.0 (0.84 ... 1.2) |
| NESM3 historical-ssp585 (1) | 30 °C | ∞ (1.1e+5 ... ∞) | 2.5 (1.9 ... 3.2) | ∞ (66 ... ∞) | 1.5 (1.3 ... 1.7) |
| NorESM2-MM historical-ssp585 (1) | 41 °C | ∞ (11 ... ∞) | 2.6 (1.3 ... 3.9) | 4.3e+7 (7.0 ... ∞) | 1.7 (1.3 ... 2.1) |
| IPSL-CM6A-LR historical-ssp585 (32) | 34 °C | ∞ (∞ ... ∞) | 2.6 (2.4 ... 2.9) | - | - |
| CAM5-1-1degree C20C historical (99) | 43 °C | 2.4e+2 (1.5e+2 ... 3.8e+2) | 1.6 (1.5 ... 1.8) | - | - |

of an unweighted average of observations and the model average. This will partly correct for a common model bias, if the observations are reliable, and is indicated by the white box accompanying the magenta bar in the synthesis figures.

Figure 12 shows the synthesis results for the current vs. past climate; the results for the future vs. current climate are presented in Figure 13. Where the results for the probability ratio do not give a finite number, we replace them by 10000 to allow all models to be included in the synthesis analysis. This means that the reported synthesised probability ratio gives a more conservative, lower value. For the intensity change we report the weighted synthesis value. For the probability ratio we can only give a lower estimate of the range. Generally, we do not see any consistent departures in the model results that can be traced back to experiment differences, except that model ensembles which consist of many members tend to have smaller uncertainties as expected.

Results for current vs. past climate, i.e., for 1.2°C of global warming vs pre-industrial conditions (1850-1900), indicate an increase in intensity of about 2.0 °C (1.2 °C to 2.8 °C) and a PR of at least 150. Model results for additional future changes if global warming reaches 2°C indicate a further increase in intensity of about 1.3 °C (0.8 °C to 1.7 °C) and a PR of at least 3, with a best estimate of 175. This means that an event like the current one, analysed here as having a return period in the current climate of 1000 years, would occur in the future world with 2°C of global warming roughly every 5 to 10 years according to the best PR estimate, albeit with large uncertainties around it. Such a 2°C climate could, according to the IPCC AR6 SSP2-4.5 which is the scenario closest to current emission levels, be reached as early as the 2040s (Lee et al., 2021).

## 6  The broader context of the heatwave

In the previous section we summarised and synthesised trends in TXx that were detected in observations and attributed to climate change using model data. In this section we provide some context to the analysed heatwave event by evaluating the assumption that this heatwave occurred in this location by chance and by discussing factors that possibly influenced the extremity of the event, being the specific meteorological conditions and dynamics, preceding dryness — which can amplify temperature during heatwaves and reduce evaporation, and the ENSO and PDO modes of of natural variability that are relevant for this region.

### 6.1  Probability of a chance event

In Section 3 we offered two explanations for having such a record breaking heatwave as that studied here; it could either occur by chance (a low-probability event) or, for instance, nonlinear effects that have not been observed at this location before could have made such an extreme heatwave possible. Here we provide some context to the first option by providing an estimate of how many heatwaves that are characterised by a return period of 1000 years at their given location we can expect on the entire globe. In this study our analysis focuses on the area 45°N-52°N, 119°W-123°W which was strongly affected by the heatwave. However, the entire area affected by heat is larger - about 1500km x 1500km which is about 1.5% of the land area of the world. Assuming that the event occurred just by chance also over the entire area affected by the heatwave and that the return period is similar to the 1000 years obtained for the study area, we can roughly estimate the global return period, i.e. the worldwide

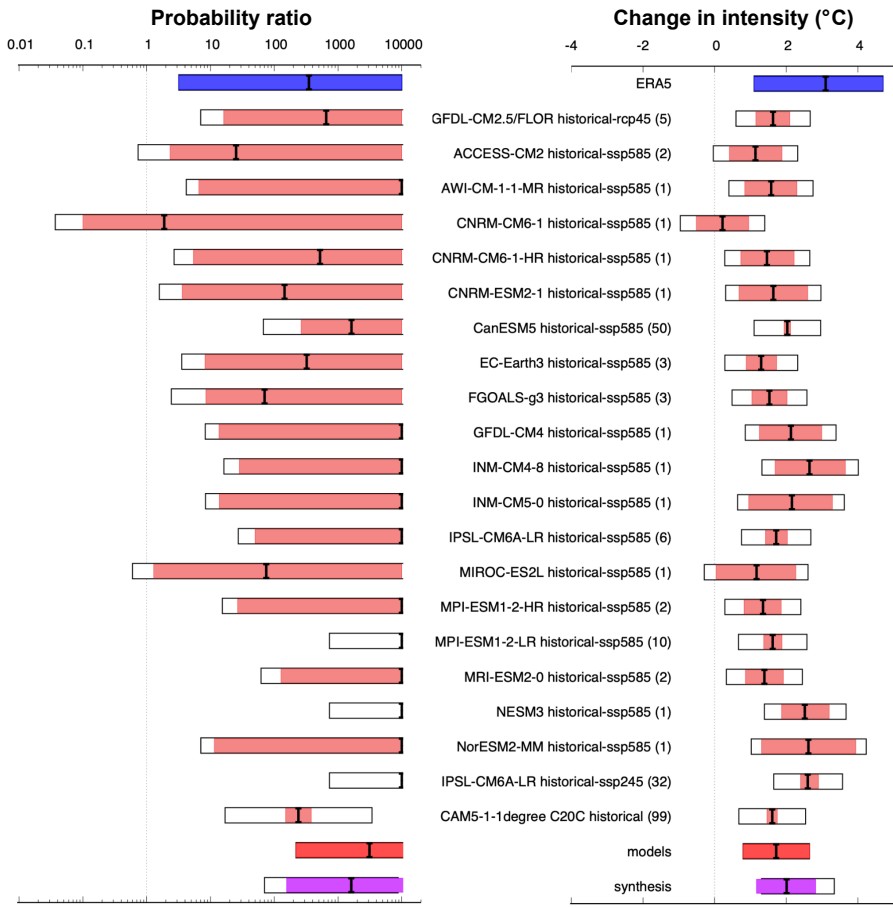

**Figure 12.** Synthesis of the past climate, showing probability ratios (left) and changes in intensity in °C (right), comparing the 2021 event with a pre-industrial climate. The blue bars show ERA5 results, the light red bars the model results, with model uncertainty shown as white bars around them. The model average is shown by the bright red bars. The magenta bars are the synthesised values, and the white box accompanying the magenta bar indicates the more conservative estimate of an unweighted average of observations and models. Model names include the model experiment and scenario and the number of ensemble members in brackets.

interval at which we would expect a heatwave similar to the one observed in terms of spatial extent and probability to occur at the given location. On the global land masses there are about $1/(1.5\%) \sim 60$ independent areas in which a heatwave of similar spatial scale could have occurred. This implies that the return period of an event as extreme as the Pacific northwest heatwave or more extreme, to occur somewhere over land, is about 60 times smaller than the estimated 1000 year interval for such an event to occur at the specific location. This gives a very rough estimate of a return period of around 15 years with a lower bound of 1.5 years (not shown) to experience such a heatwave somewhere on Earth's land area. A heatwave of this extent and extremity might therefore no longer be considered very rare if it could be expected anywhere around the globe every one or

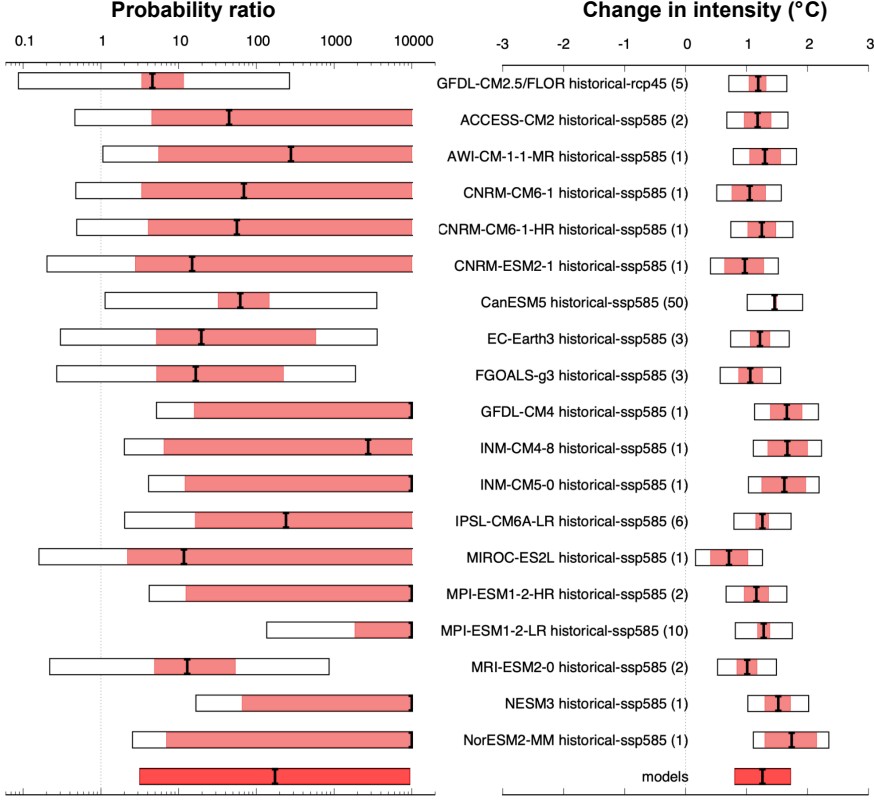

**Figure 13.** Same as Figure 12 but comparing 2°C of global warming (above pre-industrial) with present-day values (models only).

two decades. Further research on this and other exceptional heatwaves is needed to determine whether this estimate is indeed realistic, i.e., whether or not we should reject the assumption that this heatwave occurred by chance at this location.

## 6.2 Meteorological analysis and dynamics

The evolution of this event can be explained by a confluence of meso- and synoptic-scale dynamical features, potentially
including antecedent low soil moisture conditions and anomalously high column specific humidity that are a hallmarks of extreme heat in western North America (Stewart et al., 2017; Bumbaco et al., 2013). At the synoptic scale, an omega-block developed over the study area beginning at roughly 00UTC on June 25th centred at ∼125 °W, 52 °N, which then slowly progressed eastward over subsequent days. This ridge featured a maximal 500 hpa geopotential height of ∼5980 m, which is unprecedented for this area of western North America for the period from 1948 through to June 2021 at least (Figure 14).
Despite being a record, this extreme high pressure system — a feature sometimes called a "Heat dome" — is not that anomalous given the long-term trend in 500 hPa driven by thermal expansion (Christidis and Stott, 2015). Also, comparing

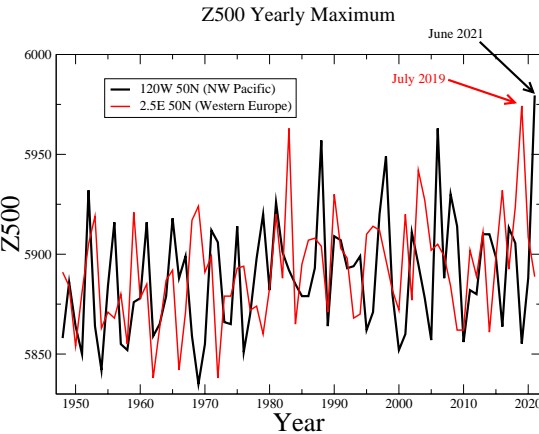

**Figure 14.** Annual maximum 500 hPa height (m) for two points at the same latitude in two continents. Black: Pacific northwest (as above) and red: Western Europe (2.5E; 50N). Data source: NCEP initialised reanalysis.

recent heatwaves in the Pacific northwest to the extreme heatwave in Western Europe in 2019 (Vautard et al., 2020), the geopotential height can be seen to reach similar anomalies and has a similar long-term trend (Figure 14).

The circulation pattern itself also appears typical for hot summertime temperatures: using analogues of 500 hPa and a
330 pattern correlation metric to compare fields, we find that about 1% of June and July circulation patterns, defined as the 500 hPa geopotential height pattern within [160 °W-110 °W; 35 °N-65 °N] in previous years have an anomaly correlation larger than 0.8 with the 28 June pattern. This degree of correlation is typical among days with this type of blocking pattern during the months of June and July. Roughly one third of June and July geopotential height fields have 1% or fewer analogues with an anomaly correlation larger than 0.8. We also find that this fraction does not change when restricting the analogues search
within 3 distinct time periods between 1948 and 2020. We conclude that the 28 June circulation is probably not exceptional, while temperatures associated with it were.

At the mesoscale, high solar irradiance during the longest days of the year and strong subsidence increased near-surface air temperatures during the event. As is typical for summer heatwaves in the region (Brewer et al., 2012, 2013), a mesoscale thermal trough developed over western Oregon by 00UTC on the 28th June. This feature migrated northward reaching the
340 northern tip of Washington State by 00UTC on the 29th. Further offshore, a small cut-off low travelled southwest to northeast around the synoptic-scale trough that made up the west arm of the omega block. The pressure gradients associated with the thermal trough and the cut-off low promoted moderate E-SE flow in the northern and eastern sectors of the feature and S-SW flow to the south. Near-surface winds with easterly components crossed the Cascade Range of Washington and Oregon and the southern Coast Mountains of British Columbia where they were lighter but sufficient to displace cooler marine air. The
345 difference in elevation on the west and east sides of the mountain ranges contributed to more adiabatic heating than cooling, which helped drive the warmest temperatures observed in the event along the foot of the west slope of these mountains, near

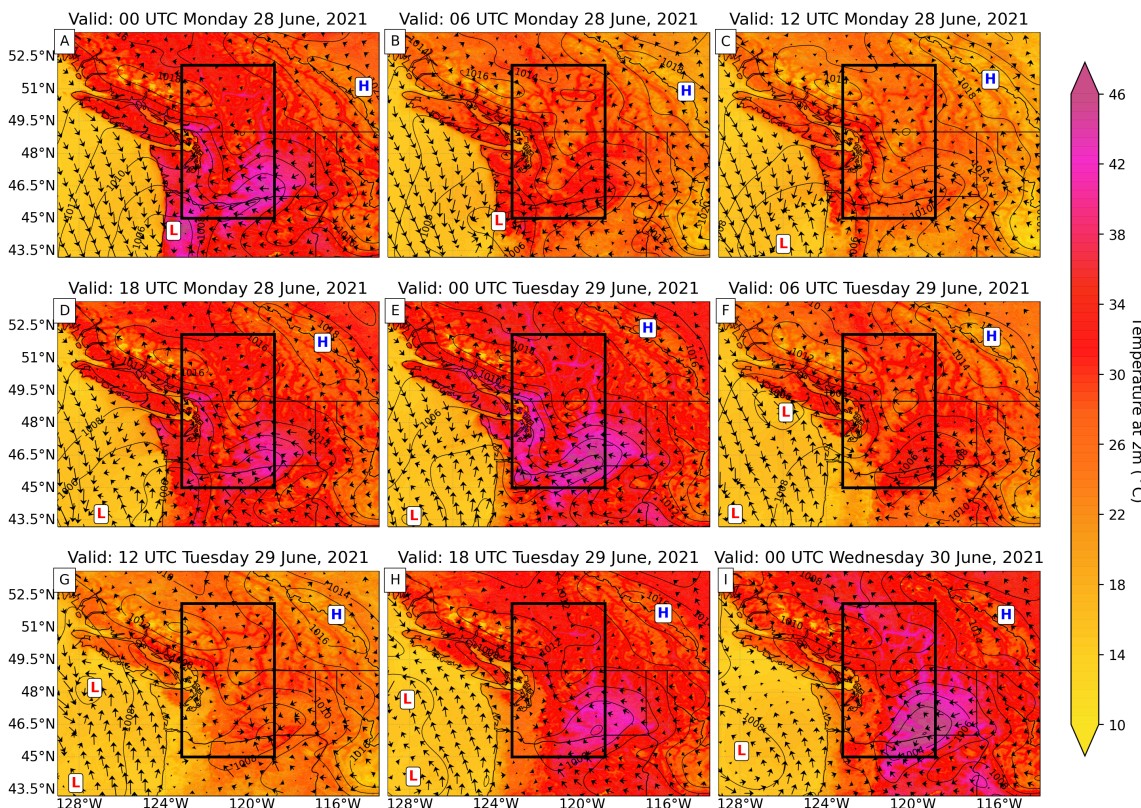

**Figure 15.** Regional simulation of sea level pressure, 2m air temperature, and 10m wind velocity in the region containing the study area (black box) using the Weather Research and Forecasting (WRF Skamarock et al., 2019) model forced by the North American Mesoscale Forecast System (NAM). Panels (A-I) show the evolution of the near-surface dynamics at a 6-hour interval from 00UTC on the 28th through 00UTC on the 30th of June 2021. Of note is Panel (E) that shows the 5PM local time on the day of peak temperature for Portland, Seattle, and Vancouver.

or slightly above sea level. These dynamics are illustrated in Figure 15. By 12UTC on the 29th of June 2021, the southwestern portion of the study area was under the influence of southerly to southwesterly near-surface flows that advected marine air and forced marked cooling. Unfortunately, winds associated with this transition intensified a wildfire that quickly consumed the town of Lytton, BC where Canada's nationwide all time high temperature was set just a day before.

There is no scientific consensus whether blocking events are made more severe or persistent because of Arctic amplification or other mechanisms (i.e. Tang et al., 2014; Barnes and Screen, 2015; Vavrus, 2018). We contend that Arctic sea ice was unlikely to have played a large role in this event largely due to the timing. In early summer, Arctic sea ice remains extensive but continues to melt, thus keeping near surface temperatures near 0 °C. This causes summer trends in near-surface temperatures

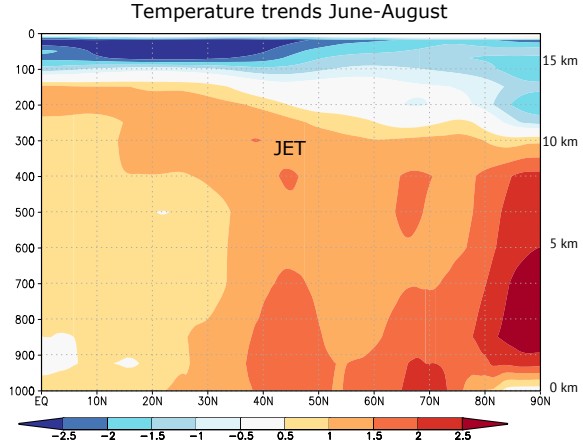

**Figure 16.** Zonal mean trends in temperature (°C per degree global warming) as a function of pressure (hPa) in the ERA5 reanalysis 1979–2019 in the northern hemisphere.

over the Arctic ocean to be smaller than for the midlatitudes. During the months prior to the event, the sea ice extent was below the 1981–2010 mean, but was similar to values observed from 2011 to 2020 (Fetterer et al., 2017 (updated daily). Instead, Arctic Amplification in summer is characterised by strong warming over high-latitude land areas (as can clearly be seen in Figure 16) and this warming signal reaches into the upper-troposphere. This enhanced warming is likely related to strong downward trends in early summer snow cover. There is evidence, from observations (Coumou et al., 2015; Chang et al., 2016), climate models (Harvey et al., 2020; Lehmann et al., 2014) and paleo-proxies (Routson et al., 2019), that this enhanced warming over high latitudes leads to a weakening of the jet and storm tracks in summer. This weakening could favour more persistent weather conditions (Pfleiderer et al., 2019; Kornhuber and Tamarin-Brodsky, 2021). Regional-scale interactions between loss of snow cover and low soil moisture associated with earlier snowmelt and rapid springtime soil moisture drying, may have had an enhanced warming impact into early summer in the Arctic. At mid-atmospheric levels there is some amplification remaining due to the winter season (Figure 16), but at the jet level ($\sim$250 hPa) the usual increase of the thermal gradient due to tropical upper tropospheric warming is advected North by the Hadley circulation (Haarsma et al., 2013). The final effect on the jet stream is therefore a competition between factors enhancing and decreasing the temperature gradient.

## 6.3 Drought

An additional feature of the event is the very dry antecedent conditions that may have contributed to the observed extreme temperatures through reduced latent cooling from inhibited evapotranspiration. Low soil moisture conditions can lead to a strong amplification of temperature during heatwaves, including nonlinear effects (Seneviratne et al., 2010; Mueller and Seneviratne, 2012; Hauser et al., 2016; Wehrli et al., 2019). In addition, low spring snow level conditions can also further amplify this feed-

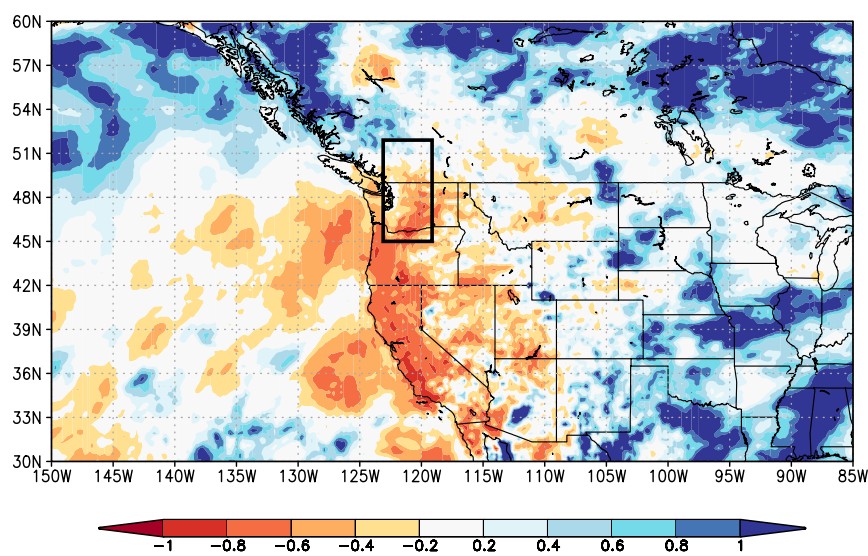

**Figure 17.** GPM/IMERG satellite estimates of relative precipitation anomalies in March–June 2021 relative to the whole record (2000–2020). The value −1 (dark red) denotes no precipitation, −0.5 (orange) 50% less than normal and zero (light grey) normal precipitation. Origin: KNMI Climate Explorer.

back (Hall et al., 2008). In this section we briefly explore whether precipitation anomalies and evapotranspiration measures could have played a role in the extreme heat via feedbacks related to soil moisture conditions.

Integrated Multi-satellitE Retrievals for the Global Precipitation Mission (IMERG) estimates of precipitation during the period from March through June, 2021 indicate anomalously dry conditions from southern BC southward through California (Figure 17). The relative precipitation anomaly ranges from close to zero over the Puget Sound area including Seattle to values of between −0.6 and −0.8, meaning that only 20%-40% of the average amount of precipitation fell in these locations, in Western Oregon. Note that in the northern parts of the area affected by the heatwave, i.e., in the coastal mountains north of

Vancouver Island, large positive precipitation anomalies occurred over the months prior to the event.

The available moisture is also influenced by evapotranspiration, which depends strongly on temperature, radiation and available atmospheric moisture. Evaporation in the study area was below normal in the ERA5 reanalysis from March and became more negatively anomalous until May (not shown). During the event in late June, there was progressive soil desiccation, creating ideal conditions for strong negative evaporation anomalies. On the other hand, surface net radiation was high, especially

before peak temperatures were reached (June 26–28). As a consequence, evaporation deficits during the heat build-up were rather slight compared to most days in May and the first half of June. Together with the extreme near-surface temperatures that

suppressed surface heating due to an already hot surface, this resulted in only moderately positive surface sensible heat flux anomalies.

Satellite-based measurements of surface soil moisture based on microwave remote sensing from the European Space Agency (ESA) Climate Change Initiative (CCI) provided by the Copernicus service suggest that surface soil moisture was below normal in the region since the beginning of April and that the anomalous conditions persisted until June (https://dataviewer.geo.tuwien.ac.at/?state=88bf0c), in agreement with the decreased precipitation and somewhat below normal evapotranspiration in the ERA5 reanalysis.

## 6.4 Influence of modes of natural variability

The El Niño Southern Oscillation is the dominant source of interannual variability in the region through the Pacific North American teleconnection. The influence is typically greatest in late winter and spring and has less clear impacts during summer and fall. Because ENSO was neutral during the preceding months and the impacts on TXx are minimal (r<0.1) we conclude that it had no influence on the occurrence of the heatwave.

The Pacific Decadal Oscillation (PDO) can affect some aspects of North American summer weather, although again the connections to heatwaves in this region are very weak. The strongly negative values of the PDO index, as they occurred in May, would slightly favour cooler conditions for this region. PDO thus also is unlikely to have played an important role in the event.

Altogether, external modes of variability appear to have played little to no role in the formation of the event.

## 7 Vulnerability and exposure

The Pacific Northwest region is not accustomed to very hot temperatures such as those experienced during the June 2021 heatwave. Heatwaves are one of the deadliest natural hazards, resulting in high excess mortality through direct impacts of heat (e.g. heat stroke) and by exacerbating pre-existing medical conditions linked to respiratory and cardiovascular issues (Haines et al., 2006; Ebi et al., 2021). In addition to at least 815 excess heat-related deaths, there was a significant increase in emergency department visits.[4] [5] On June 28, 2021 alone, there were 1,038 heat-related emergency department visits in the U.S. Department of Health and Human Services Region that includes Alaska, Idaho. Oregon, and Washington, compared with nine visits on the same date in 2019.[6] The mean daily number of heat-related illness emergency department visits in the region for 25-30 June 2021 (424) was 69-times higher than during the same days in 2019 (6). Although this region covers about 4% of the US population, it accounted for about 15% of all heat-related illness emergency department visits nationwide during June.

---

[4]https://www.nytimes.com/interactive/2021/08/11/climate/deaths-pacific-northwest-heat-wave.html

[5]https://www.cbc.ca/news/canada/british-columbia/bc-heat-dome-sudden-deaths-570-1.6122316

[6]https://www.cdc.gov/mmwr/volumes/70/wr/mm7029e1.htm

The June 2021 heatwave also affected critical infrastructure such as roads and rail and caused power outages, agricultural impacts, and forced many businesses and schools to close.[7] [8] Rapid snowmelt in BC caused water levels to rise, leading to evacuation orders north of Vancouver.[9] Furthermore, in some places, wildfires, the risk of which has increased due to climate change in this region (Kirchmeier-Young et al., 2019), started and quickly spread, requiring entire towns to evacuate.[10] The co-occurence of such events may result in compound risks, for example when households are advised to shut windows to keep outdoor wildfire smoke from getting inside while simultaneously being threatened by high indoor temperatures when lacking air conditioning.

Timely warnings were issued throughout the region by the US National Weather Service, Environment and Climate Change Canada, and local governments. British Columbia has a "Municipal Heat Response Planning" summary review that gathers information on heat response plans throughout the province, including responses such as increasing access to cooling facilities and distribution of drinking water. In long-term strategies, changes to the built environment are emphasised (Lubik et al., 2017). Not all municipalities throughout the Pacific Northwest and BC have formalised heat response plans, and others have limited planning, thought to be due to low heat risk perceptions throughout the area, as well as a lack of local data for risk assessments (Lubik et al., 2017).

The extremely high temperatures that occurred in this heat episode meant that everyone was vulnerable to its effects if exposed for a long enough period of time. Although extreme heat affects everyone, some individuals are more vulnerable, including the elderly, young children, individuals with pre-existing medical conditions, socially isolated individuals, homeless people, individuals without air-conditioning, and (outdoor) workers (Singh et al., 2019). Seattle's King County contains the third-largest population of homeless people in the U.S, with the numbers increasing during the past decade (Stringfellow and Wagle, 2018). Governmental authorities opened cooling centres throughout Seattle, Portland, and Vancouver BC during the June 2021 heatwave. [11] [12] [13] Further, electrolytes, food, and water were distributed to homeless people. [14]

The lack of air conditioning contributes to heat risk. The Pacific Northwest has lower access to air-conditioned homes and buildings compared to other regions in the U.S., with the Seattle metropolitan area being the least air-conditioned metropolitan area of the United States (<50% air conditioning in residential areas) (U.S. Census Bureau). Portland and Vancouver also have low percentages of air-conditioned households, 79% and 39% respectively (BC Hydro; U.S. Census Bureau). An increasing trend in air conditioned homes is occurring in all three cities (ibid.).

---

[7]https://apnews.com/article/canada-heat-waves-environment-and-nature-cc9d346d495caf2e245fc9ae923adae1

[8]https://www.seattletimes.com/seattle-news/weather/pacific-northwests-record-smashing-heat-wave-primes-wildfire-buckles-roads-health-toll-not-yet-known

[9]https://globalnews.ca/news/7994540/flooding-record-breaking-heat-rapid-snow-melt-bc-video/

[10]https://www.washingtonpost.com/world/2021/07/01/lytton-canada-evacuated-wildfire-heatwave/

[11]https://durkan.seattle.gov/2021/06/city-of-seattle-opens-additional-cooling-centers-and-updated-guidance-for-staying-cool-in-extreme-heat%E2%80%AF/

[12] https://www.oregonlive.com/weather/2021/06/portland-cooling-centers-provide-relief-from-heat.html

[13]https://thebcarea.com/2021/06/26/cooling-stations-set-up-around-b-c-for-record-breaking-heat-wave-this-weekend/#comments

[14]https://edition.cnn.com/2021/06/29/weather/northwest-heat-illness-emergency-room/index.html

## 8    Conclusions and recommendations

In this study, the influence of human-induced climate change on the intensity and probability of the Pacific Northwest heatwave of 2021 was investigated. We analysed how the annual maxima of daily maximum temperatures are changing with increasing global mean surface temperature, studying the area 45 °N–52 °N, 119 °W–123 °W that includes the cities Vancouver, Seattle and Portland. Synthesising results from weather observations and model simulations, we conclude that the occurrence of a heatwave of the intensity experienced in the study area would have been virtually impossible without human-caused climate change. Whilst the extremity of this event made it challenging to robustly determine how rare it was in the current climate, this general result is not strongly tied to the exact return period. For this analysis, we defined the probability of the event as one in 1000-yrs in the current climate and found that the event would have been at least 150 times rarer without human-induced climate change. Also, this heatwave was found to be about 2 °C (1.2 °C to 2.8 °C) hotter due to human induced climate change. Looking into the future to a world with 2 °C of global warming, an event like this, estimated to occur only once every 1000 years in the current climate, would occur roughly every 5 to 10 years according to the best estimate, albeit with large uncertainties around it.

This record-breaking extreme event has been analysed under the assumption that it was simply a low probability random event. A rudimentary calculation looking into the probability of a random event of similar extent and severity to occur anywhere over the earth's land area, gave an estimated chance of order 1-in-15 years, which at first impression makes a random event seem plausible, but this should be more thoroughly investigated.

The alternative is that nonlinear interactions and feedbacks occurred, which amplified the intensity of this extreme, placing it in a different population of heatwave events with different (and possibly unknown) statistics. We briefly considered dynamical and hydrological (drought) mechanisms and modes of natural variability that could have had an amplifying role. The conditions in the preceding months were dry but not extremely anomalous. The circulation itself was highly anomalous but not exceptional enough to explain the record breaking heat alone, however, local topography and preceding dryness may have amplified the associated temperatures. Also, it cannot be excluded that dynamical mechanisms (Arctic Amplification) at work influenced the persistence of blocking conditions.

Further research is planned to investigate whether these or other feedbacks were operating in this exceptional event, and whether those feedbacks are related to human-induced climate change and if they increase the frequency beyond that expected for random events of such extreme temperatures. Also, further research is needed to overcome the known limitations of standard GEV analysis on annual maxima with short records and very extreme values.

Whether or not local or dynamical feedbacks are responsible for amplifying the extreme temperatures in this particular event, this study shows that the human-induced warming that has occurred since pre-industrial conditions does make extreme events like this possible in the current climate and study region, and many times more likely than in the pre-industrial era.

Adaptation measures therefore need to be much more ambitious and take account of the rising risk of heatwaves around the world. Although this extreme heat event is still rare in today's climate, the analysis above shows that the frequency will increase with further warming. Deaths from extreme heat can be dramatically reduced with adequate preparedness action – a number

of adaptation and risk management priorities are becoming clear: it is crucial that local governments and their emergency
management partners establish heat action plans to ensure well coordinated response actions during an extreme heat event -
tailored to high-risk groups (Ebi, 2019). Heatwave early warning systems also need to be improved, which includes tailoring
messages to inform and motivate vulnerable groups, as well as providing tiered warnings that take into account vulnerable
groups may have lower thresholds for risk (Hess and Ebi, 2016). In other words, it is important to start to warn the most
vulnerable as temperatures start to rise even though the general population is not yet acutely at risk. In cases where heat action
plans and heat early warning systems are already robust, it is important that they are reviewed and updated to capture the
implications of rising risks — every five years or more often (Hess and Ebi, 2016). Further, heatwave early warning systems
should undergo stress tests to evaluate their robustness to temperature extremes beyond recent experience and to identify
modifications to ensure continued effectiveness in a changing climate (Ebi et al., 2018).

*Data availability.* Data are available via the KNMI Climate Explorer (https://climexp.knmi.nl/)

**Appendix A: Validation tables**

Table A1: As Table 1 but showing all model validation results.

| Model / *Observations* (number of members) | Seasonal cycle | Spatial pattern | Sigma | Shape parameter | Conclusion |
|---|---|---|---|---|---|
| *ERA5* | | | *1.70 (1.40 ... 1.90)* | *-0.200 (-0.500 ... 0.00)* | |
| GFDL-CM2.5/FLOR historical-rcp45 (5) | good | good | 2.01 (1.84 ... 2.17) | -0.201 (-0.272 ... -0.144) | reasonable, include as different experiment than most other models |
| ACCESS-CM2 historical-ssp585 (2) | good | good | 1.86 (1.71 ... 2.02) | -0.200 (-0.260 ... -0.120) | good |
| ACCESS-ESM1-5 historical-ssp585 (2) | good | good | 2.69 (2.49 ... 2.90) | -0.240 (-0.290 ... -0.190) | bad |
| AWI-CM-1-1-MR historical-ssp585 (1) | good | good | 1.50 (1.35 ... 1.69) | -0.200 (-0.280 ... -0.110) | good |
| BCC-CSM2-MR historical-ssp585 (1) | good | good | 2.22 (2.00 ... 2.49) | -0.230 (-0.310 ... -0.140) | bad |
| CAMS-CSM1-0 historical-ssp585 (1) | good | good | 1.98 (1.79 ... 2.23) | -0.200 (-0.290 ... -0.100) | reasonable, exclude because enough good CMIP5 models |
| CMCC-CM2-SR5 historical-ssp585 (1) | good | good | 1.29 (1.15 ... 1.46) | -0.0800 (-0.160 ... 0.0300) | reasonable, exclude because enough good CMIP5 models |
| CNRM-CM6-1 historical-ssp585 (1) | good | good | 1.54 (1.39 ... 1.72) | -0.210 (-0.290 ... -0.100) | good |
| CNRM-CM6-1-HR historical-ssp585 (1) | good | good | 1.48 (1.33 ... 1.66) | -0.190 (-0.270 ... -0.100) | good |

| | | | | | |
|---|---|---|---|---|---|
| CNRM-ESM2-1 historical-ssp585 (1) | good | good | 1.71 (1.54 ... 1.92) | -0.180 (-0.250 ... -0.0900) | good |
| CanESM5 historical-ssp585 (50) | good | reasonable | 1.79 (1.76 ... 1.82) | -0.180 (-0.190 ... -0.170) | reasonable, include because statistical parameters good |
| EC-Earth3 historical-ssp585 (3) | good | good | 1.87 (1.76 ... 2.00) | -0.220 (-0.270 ... -0.170) | good |
| EC-Earth3-Veg historical-ssp585 (4) | good | good | 2.07 (1.95 ... 2.19) | -0.250 (-0.290 ... -0.210) | bad |
| FGOALS-g3 historical-ssp585 (3) | good | reasonable | 1.80 (1.69 ... 1.92) | -0.180 (-0.210 ... -0.140) | reasonable, include because statistical parameters good |
| GFDL-CM4 historical-ssp585 (1) | good | good | 1.43 (1.29 ... 1.62) | -0.210 (-0.300 ... -0.110) | good |
| GFDL-ESM4 historical-ssp585 (1) | good | good | 1.37 (1.23 ... 1.55) | -0.170 (-0.260 ... -0.0700) | reasonable, exclude because enough good CMIP5 models |
| HadGEM3-GC31-LL historical-ssp585 (4) | good | good | 2.00 (1.90 ... 2.12) | -0.210 (-0.250 ... -0.170) | reasonable, exclude because enough good CMIP5 models |
| HadGEM3-GC31-MM historical-ssp585 (3) | good | good | 2.08 (1.96 ... 2.22) | -0.190 (-0.230 ... -0.140) | bad |
| INM-CM4-8 historical-ssp585 (1) | good | good | 1.63 (1.46 ... 1.83) | -0.210 (-0.300 ... -0.110) | good |
| INM-CM5-0 historical-ssp585 (1) | good | good | 1.80 (1.63 ... 2.03) | -0.240 (-0.310 ... -0.140) | good |
| IPSL-CM6A-LR historical-ssp585 (6) | good | reasonable | 1.79 (1.71 ... 1.88) | -0.220 (-0.250 ... -0.180) | reasonable, include because statistical parameters good |
| KACE-1-0-G historical-ssp585 (3) | good | good | 2.27 (2.13 ... 2.41) | -0.241 (-0.282 ... -0.196) | bad |
| MIROC-ES2L historical-ssp585 (1) | reasonable, peaks about a month early | reasonable | 1.46 (1.31 ... 1.65) | -0.190 (-0.300 ... -0.0900) | reasonable, include because statistical parameters good |
| MIROC6 historical-ssp585 (50) | good | good | 1.31 (1.29 ... 1.33) | -0.220 (-0.220 ... -0.210) | bad |
| MPI-ESM1-2-HR historical-ssp585 (2) | good | good | 1.49 (1.39 ... 1.62) | -0.250 (-0.310 ... -0.190) | good |
| MPI-ESM1-2-LR historical-ssp585 (10) | good | good | 1.63 (1.58 ... 1.69) | -0.260 (-0.280 ... -0.230) | good |
| MRI-ESM2-0 historical-ssp585 (2) | reasonable, peak too flat | good | 1.41 (1.30 ... 1.53) | -0.280 (-0.340 ... -0.220) | reasonable, include because statistical parameters good |
| NESM3 historical-ssp585 (1) | good | good | 1.48 (1.34 ... 1.67) | -0.290 (-0.370 ... -0.200) | good |
| NorESM2-MM historical-ssp585 (1) | good | good | 1.90 (1.70 ... 2.12) | -0.250 (-0.350 ... -0.140) | in between reasonable and good, include |

| | | | | | |
|---|---|---|---|---|---|
| UKESM1-0-LL historical-ssp585 (5) | good | good | 1.99 (1.90 ... 2.09) | -0.170 (-0.190 ... -0.140) | reasonable, exclude because enough good CMIP5 models |
| IPSL-CM6A-LR historical-ssp245 (32) | good, from CMIP6 | reasonable, from CMIP6 | 1.69 (1.64 ... 1.75) | -0.220 (-0.250 ... -0.200) | reasonable, observed GMST used, include |
| GFDL-AM2.5C360 historical (5) | good | good | 2.15 (1.99 ... 2.30) | -0.259 (-0.335 ... -0.197) | bad, variability too high |
| CAM5-1-1degree C20C historical (99) | NA | NA | 1.70 (1.68 ... 1.72) | -0.176 (-0.172 ... -0.180) | good, values used with warming level 1.7 |
| MIROC5 C20C historical () | NA | NA | 1.36 (1.33 ... 1.39) | -0.240 (-0.224 ... -0.256) | bad |
| HadGEM3-A-N216 C20C historical () | NA | NA | 2.00 (1.95 ... 2.05) | -0.240 (-0.218 ... -0.262) | bad |

*Author contributions.* SYP and SFK are the study co-leads, GJvO, FSA, SIS, RV, DC, KLE, JA, RS, MvA and CPM provided analyses and text, MW, WY, SL, DLS, MH, RB and LNL provided data and the synthesis, FL, NG, JT and GAV improved the text, CR, RBS and RH provided figures and FELO initiated the study and contributed to the discussions.

*Competing interests.* The authors declare no competing interest.

*Acknowledgements.* We acknowledge the World Climate Research Programme, which, through its Working Group on Coupled Modelling, coordinated and promoted CMIP6. We thank the climate modeling groups for producing and making available their model output, the Earth System Grid Federation (ESGF) for archiving the data and providing access, and the multiple funding agencies who support CMIP6 and ESGF. We thank Urs Beyerle for downloading and curating the CMIP6 data at ETH Zurich. F.L. was supported by the Regional and Global Model Analysis (RGMA) component of the Earth and Environmental System Modeling Program of the U.S. Department of Energy's Office 495 of Biological & Environmental Research (BER) via National Science Foundation IA 1947282.

We would like to express our deep gratitude and respect to Geert Jan van Oldenborgh, co-author of this manuscript, who sadly passed away on 12th October 2021. His outstanding contribution to the science of extreme event attribution and his passion to provide the knowledge, data and analysis tools (https://climexp.knmi.nl/) that are most relevant for society and especially for vulnerable communities has been enormous. We have lost an incredible scientist, passionate colleague, and friend.

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
