# Peer review of "Rapid attribution analysis of the extraordinary heatwave on the Pacific Coast of the US and Canada June 2021"

_Earth System Dynamics, 2021_

## Referee Comment (RC2)

Reviewer's comment on the manuscript
"Rapid attribution analysis of the extraordinary heatwave on the Pacific Coast of the US and Canada June 2021" Philip Sjoukje et al.

General comments

The authors try to assess in this manuscript to what extent the June 2021 heatwave affecting the Pacific Northwestern coast of US and Western Canada is attributable to climate change. They use multi-method multi-model attribution, and rely on the output of several (around 20) earth system models, reanalysis data, and meteorological observations. They also analyse qualitatively the meteorological conditions before and during the event, and discuss aspects of vulnerability and exposure.

Except the relevance of the topic, I don't see any reason for the publication of this manuscript. The quality of the study and of the manuscript is not good enough for publication in a peer reviewed scientific journal. In the following I summarise my main concerns.

The scientific methods are not adequate.

The Generalized Extreme Value (GEV) distribution is a limiting distribution. This means that the block size has to be large enough to allow for the convergence of the estimated GEV distribution to the asymptotic GEV. This should be checked based on the convergence of the shape parameter as function of increasing block size (see for example Coles, 2001, Springer). **Annual maxima are not automatically GEV distributed**. In fact, looking for example at Fig. 6b, it looks like the curvature of the fitted line would change if one would start the fit at larger return periods (e.g. 5 yr or 10 yr. This suggests that the shape parameter is not converged in case of annual maxima, thus the extrapolation towards unobserved events is not underpinned by the theory. One needs to select maxima over larger blocks, assuming that there is a convergence at all. However, if we would take for example block sizes of 10 years, we would not have enough maxima to estimate the GEV parameters properly, furthermore the uncertainty would substantially increase. Thus, I'm afraid that larger blocks will not solve the problem either, because of the lack of sufficient data. The slow convergence and the "waste" of data by considering only block maxima are common application issues in case of this method.

Now, let's leave the convergence issue beside and concentrate on the GEV fit. Fig 6b, 7b, and 8b show:
1) if one omits the 2021 value, the fit seems to be appropriate for the rest of the annual maxima but the 2021 value is substantially above the upper limit of the estimated distribution (Fig. 6b);
2) things don't get better if one considers the 2021 event as well, the fit is much worse generally and the 2021 event still lies outside the confidence intervals (Fig. 8b).

This can have following reasons:
1) the 2021 event belongs to a different parent distribution – this would be a violation of the condition of identically distributed variables, as required by the extreme value law;
2) the empirical return level is underestimated and the event is much rarer then it seems. Indeed, the maxima of a 72 years long times series can be a 1 in 72 years event, but also a 1 one in 100 or maybe a 1 in 1000 year event or even rarer.

The authors account for nonstationarity of extremes only based on a linear trend in the location parameter. However, there might be other sources of non-stationarity as well, leading to unrealistic estimates.

I don't understand at all the point of the method used to produce Fig. 7b. The authors write that "we still assume that the data up to 2020 can be described by a GEV with constant scale and shape parameters, but we reject all GEV models in which the upper bound is below the value observed in 2021". You have one time series based on which you estimate the GEV parameters and their confidence intervals. Where do all these GEV models come from? Besides, the advantage of fitting a GEV distribution is that the data itself "decides" about the optimal parameters. The way you proceed, you lose this advantage and it seems like you take a subjective decision to obtain specific results.

**Based on the above reasons, I think that the block maxima method at the selected block size does not provide a reliable statistical model for the analysed temperature extremes, thus the conclusions of the manuscript are not reliable as well.**

I also have concerns related to the definition of the heatwaves. Actually by taking the annual maxima we define the event at one point in time, thus we lose the information about the duration of the event, which is crucial in case of persistent extreme events, like heatwaves. Heatwaves with very different duration, let's say 3 days vs. 3 weeks, could produce the same annual maxima, and still be caused by different circulation anomalies and physical processes. However, the used definition cannot differentiate between several types of events. Another issue is that the annual maxima are then averaged over the study area. But we don't know if the maxima at different grid points refer to the same event. It is indeed very probable that they do not and in that case we would compute spatial averages of events happening at different time steps, which makes no sense.

The results provide almost no valuable information, thus the conclusions make no sense.

Considering the inadequate methodology and the huge uncertainties in the return periods and probability ratios, I'm afraid that the main conclusions are not trustworthy.

The authors do not state clearly which GEV fit they follow, but based on the final conclusion of "1 in 1000 year" event, I assume that they decide for the one shown in Fig. 8b. One of the final conclusions is that "the event is estimated to be about a 1 in 1000 year event in today's climate". I point out that one cannot obtain correct return level estimates for an event occurring on average once every 1000 years based on **72** values of **annual** maxima.

Another main conclusion is that "an event … as rare as 1 in a 1000 years would have been at least 150 times rarer without human-induced climate change", meaning that the uncertainty bounds for the probability ratio are 150 and infinity. And finally "in a world with 2 °C of global warming … a 1000-year event … would occur roughly every 5 to 10 years". This last statement is obtain based on the best estimate of the probability ratio of 175, with the uncertainty ranging from **3 to infinity**, giving a return period range of from 1 year to 333 years. There is too much uncertainty in these results making them not very useful.

I am puzzled as well by the infinite probability ratios. I assume that they come because of division by 0, which would be problematic since it does not say anything about the magnitude of the numerator.

However, the authors do not define the probability ratio in the paper, thus one can just assume that this is the case.

The language is not clear enough, the formulation not precise enough, and the structure confusing.

The quality of the scientific text is not good enough, and for me it is very strange that it was submitted in this form, considering also the number of authors of this paper.

There are many sentences/paragraphs where I don't get the message mainly because of wrong and/or inaccurate phrasing. Section 2, 3, 6 are particularly critical. I do not understand the meaning of Sec. 7.1 at all, and sentences like "This implies that the return time of an event as rare as this one or rarer, somewhere over land, is about 60 times smaller than the O(1000 yr) that it occurred at the specific location that it did." do not help.

The quality of many figures is not good enough:

- Several figures are direct output of a climate explorer tool with a poor choice of colours, and confusing axis labels. For example, what is "max Tmax"? Tmax is used also in the text, however it is not defined. It took me a while to find out the difference between TXx and Tmax. It is very confusing and it is not explained.
- Fig. 12 and Fig 13 are not self-explanatory. The meaning of the colours is explained in the text, but not in the figure caption.
- Fig 12 shows the synthesis results of the **past** climate compared with **today's** climate. Why is "historical-ssp" written in the figure legend?
- Fig. 15: I do not see the circulation changes described in the text. Assuming that the study area is the same as defined at the beginning of the manuscript, I cannot see at all a "southerly to southwesterly near surface flow" in the study area in panel B.
- Fig. 17: The figure label contains: "Fit: KNMI Climate Explorer". What is fitted in this case?
- In case of the maps, it would be good to mark the study area.

The structure of the paper is confusing:

- There are many sections and some of them are very short. For example, section 5 has 4 lines. It could be incorporated in section 6.
- The paper has no summary joining the main results from the different sections. The different topics (for example, the "statistical estimation", "synthesis results", "meteorological analysis") are like puzzle pieces which are not merged at the end.
- Line 136: "As discussed in section 1.2, we analyse the annual maximum of daily maximum temperatures (TXx)". This is discussed in section 1.1, actually.
- You mention all those different experiments in Sec. 2.2. The CM6A ensemble "is used to explore the influence of climate variability", you use high resolution GFDL and AMIP runs, but you do not explain what we learn from these runs, you just include them, together with all the other models, in the synthesis analysis.
- The second part of the abstract is not clear and precise enough.
- The probability ratio PR is not defined.

---

## Author Comment (AC2)

Reply to reviewer's comment on the manuscript "Rapid attribution analysis of the extraordinary heatwave on the Pacific Coast of the US and Canada June 2021" Philip Sjoukje et al.

*Our replies to the reviewer comments are in italic.*

**General comments**
The authors try to assess in this manuscript to what extent the June 2021 heatwave affecting the Pacific Northwestern coast of US and Western Canada is attributable to climate change. They use multi-method multi-model attribution, and rely on the output of several (around 20) earth system models, reanalysis data, and meteorological observations. They also analyse qualitatively the meteorological conditions before and during the event, and discuss aspects of vulnerability and exposure.

Except the relevance of the topic, I don't see any reason for the publication of this manuscript. The quality of the study and of the manuscript is not good enough for publication in a peer reviewed scientific journal. In the following I summarise my main concerns.

*We thank the reviewer for the feedback and for taking the time to review our manuscript. We are fully aware of the issues raised by the reviewer and try to better discuss them in the revised version, and make clear the assumptions we make while still applying standard methods, which have been applied to many other cases. Indeed, the study is based on peer-reviewed methods which have been tested repeatedly in different contexts and also highlighted by the most recent IPCC WG1 report as an important new scientific development. The report resulting from the initial rapid study was published on the World Weather Attribution website. As, however, this heat event showed some peculiar characteristics, such as the fact that its severity is lying well above the expected range making the fitting of a distribution challenging, we feel it is important to publish this analysis in peer-reviewed literature using these methods as in previous studies, while mentioning the potential limits of this standard approach. This study also raises many interesting questions and will form the foundation for further studies of the analysed heatwave by members of the team and by other colleagues. Finally the study not only uses observations, for which the assessment of the fit is difficult to make, but also uses simulations, allowing to explore larger datasets, and includes a discussion combining both lines of evidence.*
*In the following, we will address the concerns and implement improvements into the manuscript.*

**The scientific methods are not adequate.**
The Generalized Extreme Value (GEV) distribution is a limiting distribution. This means that the block size has to be large enough to allow for the convergence of the estimated GEV distribution to the asymptotic GEV. This should be checked based on the convergence of the shape parameter as function of increasing block size (see for example Coles, 2001, Springer). **Annual maxima are not automatically GEV distributed**. In fact, looking for example at Fig. 6b, it looks like the curvature of the fitted line would change if one would start the fit at larger return periods (e.g. 5 yr or 10 yr. This suggests that the shape parameter is not converged in case of annual maxima, thus the extrapolation towards unobserved events is not underpinned by the theory. One needs to select maxima over larger blocks, assuming that there is a convergence at all. However, if we would take for example block sizes of 10 years, we would not have enough maxima to estimate the GEV parameters properly, furthermore the uncertainty would substantially increase. Thus, I'm afraid that larger blocks will not solve the problem either, because of the lack of sufficient data. The slow convergence and the "waste" of data by considering only block maxima are common application issues in case of this method.

Now, let's leave the convergence issue beside and concentrate on the GEV fit. Fig 6b, 7b, and 8b show:
1) if one omits the 2021 value, the fit seems to be appropriate for the rest of the annual maxima but the 2021 value is substantially above the upper limit of the estimated distribution (Fig. 6b);
2) things don't get better if one considers the 2021 event as well, the fit is much worse generally and the 2021 event still lies outside the confidence intervals (Fig. 8b).

This can have following reasons:
1) the 2021 event belongs to a different parent distribution – this would be a violation of the condition of identically distributed variables, as required by the extreme value law;
2) the empirical return level is underestimated and the event is much rarer then it seems. Indeed, the maxima of a 72 years long times series can be a 1 in 72 years event, but also a 1 one in 100 or maybe a 1 in 1000 year event or even rarer.

The authors account for nonstationarity of extremes only based on a linear trend in the location parameter. However, there might be other sources of non-stationarity as well, leading to unrealistic estimates.

I don't understand at all the point of the method used to produce Fig. 7b. The authors write that "we still assume that the data up to 2020 can be described by a GEV with constant scale and shape parameters, but we reject all GEV models in which the upper bound is below the value observed in 2021". You have one time series based on which you estimate the GEV parameters and their confidence intervals. Where do all these GEV models come from? Besides, the advantage of fitting a GEV distribution is that the data itself "decides" about the optimal parameters. The way you proceed, you lose this advantage and it seems like you take a subjective decision to obtain specific results.

**Based on the above reasons, I think that the block maxima method at the selected block size does not provide a reliable statistical model for the analysed temperature extremes, thus the conclusions of the manuscript are not reliable as well.**

*The reviewer identified the reason for why this study was so difficult, given the extremeness of the event and the relatively small number of samples, but we see this as a strong argument to publish this manuscript in a peer-reviewed journal, provided we recognize these difficulties. We use peer reviewed methods (Philip et al., 2020 and Van Oldenborgh et al., 2021) for our analyses, which have proven to be useful in many heatwave analyses (e.g., Kew et al 2019, Vautard et al. 2020, Ciavarella et al. 2021). In this case, , the event is so extreme that the assumptions underlying the GEV fit are hard to assess and the uncertainty itself is very difficult to estimate. In addition, other mechanisms may have been playing a role. We illustrated this in figures 6-8.*

*Furthermore we expanded the text, clarifying the options and decisions:*
- *We added an explanation in the text on the GEV fit that excludes the event:*
  *"The GEV fit has a negative shape parameter ξ, which implies a finite tail, and hence an upper bound, here it is at 35.5±1.3 ℃ (2σ uncertainty). However, the value observed in 2021, 39.7 ℃, is far above this upper bound. Therefore, this GEV fit with constant shape and scale parameters that excludes all information about 2021 does not provide a valid description of heatwaves in the area."*

- *Next, we improved the wording of the second option:*
  *"An alternative to the standard approach for which no information of the event itself is used (to avoid a selection bias), is to use the information that it actually happened, yet without including the value observed in 2021 in the fit. Specifically, we still assume that the data up to 2020 can be described by a GEV distribution with constant scale and shape parameters, but we reject all GEV models in which the upper bound is below the value observed in 2021. In other words, we enforce that the fit parameters are within a subset of parameters that are compatible with the 2021 event."*

- *On the third approach, acknowledging the difficulties with the estimation of return periods of this event, we conclude that "Given the extremity of the event and the relatively low number of data, a robust GEV fit is hard to obtain, and the appropriateness of the method is difficult to assess. However, the application of this classical method in this case is interesting provided we keep in mind the assumptions we make. While we acknowledge that none of the three possibilities to fit a GEV distribution is fully satisfying, we decided on using this third approach to estimate the return period leading to an estimate of 1,000 years (95% CI >100 yr). Follow-up research will be necessary to investigate the potential reasons for this outstanding event and the consequences on assumptions for these fits (see also the discussion in Sections 6 and 8)."*
  *Please note that Section 6 and 8 are the sections on the broader context and the conclusion, which were the old Sections 7 and 9.*

*It is very important to publish the analysis results for this heatwave event and, especially due to the discussed challenges with fitting a distribution to the data, to point out open research questions arising through analysis of this event that need more in depth research. We show and acknowledge the limitations of the approach, and we now conclude in Section 6.1 (old Section 7.1): "Further research on this and other exceptional heatwaves is needed to determine whether this estimate is indeed realistic, i.e., whether or not we should reject the assumption that this heatwave occurred by chance at this location.".*
*And also related to this issue we add to Section 3.2:*

*"further research is needed into the limitations of standard GEV analysis on annual maxima with short records and seemingly non-stationary behavior. Climate model large ensembles offer a future test bed to investigate the robustness of the method in light of current limitations."*

*In agreement with the reviewers comments, extreme weather attribution would become easier if we would have much more data, including many more extremes. In that case we could use block sizes of 10 years (or even better, 20 or 30 years), but as observed data sets expand slowly - day by day and year by year, this is not the case now and will not be the case soon.As always, a balance has to be found between using as much of the data as possible and also characterizing the behavior of the tail adequately and thus the extremity of an event. We therefore argue that we used the available data appropriately and sharpened the discussion of initial possibilities and issues in the manuscript, see the changes listed above.  We also tested other block lengths (2 and 5 years), and found similar results as for 1 year, which at least provides consistency, even though it does not allow a full assessment of the validity of underlying assumptions. Finally, we do not implicitly include nonlinearities and nonstationarity other than the linear trend wrt GMST, as explained in the manuscript:*
*"The second option is that strong nonlinear interactions and feedbacks took place in this event with yet unseen temperatures, relative to previous heatwaves, amplifying the intensity of the extreme, a sign that climate change could exacerbate extreme heatwaves beyond expected temperatures. In this case, the event would not belong to the ''same population'' as the other ones and we would not expect the method applied here to be successful. This second possibility requires further investigation. We keep this possibility in mind, along this study, but we use the first assumption."*

*Following World Weather Attribution's peer-reviewed method, we decide on the best estimate of the return period and use that return period in the model analysis, even though the technique used to estimate the return period may be not ideal due to issues raised in the paper and by the reviewer. This means, as explicitly stated in the paper, that regardless of the exact return time of the observed event, the results and the synthesis are valid for the model analysis at that return period and thus provide a valuable assessment of very rare heat extremes in a changing climate in the region. For model data, the GEV is a good fit for TXx when we use the same peer-reviewed method and the GEV shifting with GMST. The results and conclusions are thus valuable and valid in a general sense: the frequency of (as measured in the current climate) 1-in-1000 year events in this region has increased dramatically with rising GMST. We argue that an attribution statement using this estimate of the return period is much more valuable than no attribution statement, as long as the limitations are discussed as well, which we do mainly in section 3.1.*

*Concerning the point on figure 7b: A full description of the statistical methods is given in Philip et al. 2020 and Van Oldenborgh et al. 2021. In the GEV fit, the confidence intervals are estimated using a non-parametric bootstrap, so the fit is done many times using samples of covariate-observations pairs drawn from the original series with replacement. In case a GEV fit indicates the event observed in 2021 would not be possible, i.e.the upper bound is below the value observed in 2021, we do not include it in the final distribution. In other words, we use the prior knowledge that the event occurred by demanding that the distribution fitted to any bootstrapped sample has a non-zero probability of the observed event occurring. Of course we lose the advantage of fitting the optimal parameters, but instead account for the fact that we already know that the optimal parameters would not adequately represent the true distribution, given the event happened. This step is deliberately a compromise between excluding and including the event in the fit, and thus not common practice but aimed to use as much information as possible without prescribing the answer. This is now included more clearly in the text (see listed text above).*

*references:*
*Ciavarella, A., Cotterill, D., Stott, P. et al. Prolonged Siberian heat of 2020 almost impossible without human influence.Climatic Change **166**, 9 (2021). https://doi.org/10.1007/s10584-021-03052-w*

*Kew S F, Philip S Y, Jan van Oldenborgh G, van der Schrier G, Otto F E and Vautard R 2019 The exceptional summer heat wave in southern Europe 2017 Bull. Am. Meteorol. Soc. **100** S49-S53.*

*Philip, S. Y., Kew, S. F., van Oldenborgh, G. J., Otto, F. E. L., Vautard, R., van der Wiel, K., King, A. D., Lott, F. C., Arrighi, J., Singh, R. P., and van Aalst, M. K.: A protocol for probabilistic extreme event attribution analyses, Advances in Statistical Climatology, Meteorology and Oceanography, 6, 177–203, https://doi.org/10.5194/ascmo-6-177-2020, 2020.*

*van Oldenborgh, G. J., van der Wiel, K., Kew, S., Philip, S., Otto, F., Vautard, R., King, A., Lott, F., Arrighi, J., Singh, R., and van Aalst, M.: Pathways and pitfalls in extreme event attribution, Climatic Change, 166, 13, https://doi.org/10.1007/s10584-021-03071-7, 2021.*

*Vautard R et al (2020) Human contribution to the record-breaking June and July 2019 heatwaves in Western Europe. Environ Res Lett 15(9)*

I also have concerns related to the definition of the heatwaves. Actually by taking the annual maxima we define the event at one point in time, thus we lose the information about the duration of the event, which is crucial in case of persistent extreme events, like heatwaves. Heatwaves with very different duration, let's say 3 days vs. 3 weeks, could produce the same annual maxima, and still be caused by different circulation anomalies and physical processes. However, the used definition cannot differentiate between several types of events. Another issue is that the annual maxima are then averaged over the study area. But we don't know if the maxima at different grid points refer to the same event. It is indeed very probable that they do not and in that case we would compute spatial averages of events happening at different time steps, which makes no sense.

*The reviewer is right that we do not distinguish between events with different duration. The first reason to study this event is that it was an exceptional heat event, and heat extremes both short and long have an impact on human health Vicedo-Cabrera (2021) and Guo (2017) and infrastructure. At the event definition stage, we choose spatial and temporal definitions that represent the event from the viewpoint of impacts, not necessarily the meteorological conditions. In this case, we chose the record breaking maximum temperature rather than the duration of heat above a threshold because this is what characterised this exceptional out-of-the-range event. In the section on "Event definition" we explain the reasoning behind the choice for TXx. We indeed do not distinguish between different circulation types and physical processes. Of course that could be done, following the storyline approach to attribution but the storyline approach answers a different question: in the current analysis we answered whether and to what extent a heatwave as strong as in June 2021 over the PNA area has been altered by human induced climate change. Although we would like to expand to the reasoning behind the extremity in a further study, this was not the scope of the current analysis.*
*Furthermore, we first average over the area and only afterwards select TXx. Therefore, TXx only includes data from one individual day. We now explicitly mention this in the manuscript:*
*"Here, we first average over the region and then take the annual maximum."*

*References:*
*Vicedo-Cabrera AM and co-authors. The burden of heat-related mortality attributable to recent human-induced climate change. Nat Clim Chang. 2021 Jun;11(6):492-500. doi: 10.1038/s41558-021-01058-x. Epub 2021 May 31. PMID: 34221128; PMCID: PMC7611104.*

*Guo Y and co-authors. Heat Wave and Mortality: A Multicountry, Multicommunity Study. Environ Health Perspect. 2017 Aug 10;125(8):087006. doi: 10.1289/EHP1026. PMID: 28886602; PMCID: PMC5783630.*

The results provide almost no valuable information, thus the conclusions make no sense.

Considering the inadequate methodology and the huge uncertainties in the return periods and probability ratios, I'm afraid that the main conclusions are not trustworthy.

*We politely disagree with the reviewer. Although there are huge uncertainties in the return periods and probability ratios, the results have already proven to be valuable, and we clearly discuss the assumptions made. Even though the uncertainty in the probability ratio is large, we report the very important conclusion that the probability ratio is far above 1 (>150). While this is in itself not surprising, the finding that the observed data does not allow to estimate the return time with confidence is in itself important, highlighting that prevention measures currently in place cannot rely on observations alone to adequately prepare the population in a rapidly changing climate. Even if the "true" return time of the event is higher or lower, as explained above, the observational analysis gives important insights for a 1-in-1000 year event, a benchmark used by many different stakeholders. This information is very valuable for decision makers and in informing the general public, and much more informative than a statement such as 'we don't know anything because the event was too extreme'. Our findings also raise a number of questions for future research, in particular about the potential of nonlinearities. The paper is linking a concrete experience with current scientific*

*understanding of heat extremes in a changing climate, demonstrating how much return periods change for very rare events and highlighting where there is need for further research.*

The authors do not state clearly which GEV fit they follow, but based on the final conclusion of "1 in 1000 year" event, I assume that they decide for the one shown in Fig. 8b. One of the final conclusions is that "the event is estimated to be about a 1 in 1000 year event in today's climate". I point out that one cannot obtain correct return level estimates for an event occurring on average once every 1000 years based on **72** values of **annual** maxima.

Another main conclusion is that "an event … as rare as 1 in a 1000 years would have been at least 150 times rarer without human-induced climate change", meaning that the uncertainty bounds for the probability ratio are 150 and infinity. And finally "in a world with 2 °C of global warming … a 1000-year event … would occur roughly every 5 to 10 years". This last statement is obtain based on the best estimate of the probability ratio of 175, with the uncertainty ranging from **3 to infinity**, giving a return period range of from 1 year to 333 years. There is too much uncertainty in these results making them not very useful.

I am puzzled as well by the infinite probability ratios. I assume that they come because of division by 0, which would be problematic since it does not say anything about the magnitude of the numerator.
However, the authors do not define the probability ratio in the paper, thus one can just assume that this is the case.

*We think the understanding of what choices we made for the analysis and why we opted for them is very important and therefore we tried to communicate that the choice of the GEV fit does influence the reported numbers. Therefore, in the abstract we stated that:*
*"This makes it hard to quantify with confidence how rare the event was. Using a statistical analysis that assumes that the heatwave is part of the same distribution as previous heatwaves, i.e. it can be included in the statistical fit and it was not intensified by new nonlinearities, a first order estimation of the event frequency is of the order of once in 1000 years under current climate conditions. ."*

*The reason for the infinities is, as the reviewer correctly noticed, because the event has zero probability in the past. This is exactly why we do not only report probability ratios but also intensity changes, which are much better confined for extreme heatwave events.*
*We agree that there are large uncertainties involved in the estimate of the return period for the future, but still think it is worth mentioning a number to emphasize the urgency of adapting to our changing climate. We now emphasise that these estimates are surrounded by large uncertainties. The text now reads:*
*"Model results for additional future changes if global warming reaches 2 °C indicate a further increase in intensity of about 1.3 °C (0.8 °C to 1.7 °C) and a PR of at least 3, with a best estimate of 175. This means that an event like the current one, analysed here as having a return period in the current climate of 1000 years, would occur in the future world with 2 °C of global warming roughly every 5 to 10 years according to the best PR estimate, albeit with large uncertainties around it."*

*We are sorry we did not state the definition of the probability ratio clearly, although it is contained in the background literature Philip et al., 2020 and Van Oldenborgh et al., 2021. We will add the definition to the manuscript.*

The language is not clear enough, the formulation not precise enough, and the structure confusing.

The quality of the scientific text is not good enough, and for me it is very strange that it was submitted in this form, considering also the number of authors of this paper.

There are many sentences/paragraphs where I don't get the message mainly because of wrong and/or inaccurate phrasing. Section 2, 3, 6 are particularly critical. I do not understand the meaning of Sec. 7.1 at all, and sentences like "This implies that the return time of an event as rare as this one or rarer, somewhere over land, is about 60 times smaller than the O(1000 yr) that it occurred at the specific location that it did." do not help.

*Thank you very much for pointing out some of the deficiencies in the wording of our manuscript. To improve the text, we have started working through the entire manuscript to ensure our results are understandable to a wide audience. In Section 7.1 (which now became section 6.1) we provide an estimate of the probability to have a similarly extreme event anywhere on the land surface of the globe. We have now reworded this section as follows:*
*"In this study our analysis focuses on the area 45°N-52° N, 119° W-123°W which has been strongly affected by the heatwave. However, the entire area affected by heat is larger - about 1500km x 1500km, which is about 1.5% of the*

*land area of the world. Assuming that the event occurred just by chance, we can roughly estimate the global return time, i.e. the worldwide interval at which we would expect a heatwave similar to the one observed in terms of spatial extent and probability to occur at the given location. On the global land masses there are about 1/(1.5%) ~ 60 independent areas in which a heatwave of similar spatial resolution could have occurred. This implies that the return time of an event as rare as the Pacific Northwest heatwave or rarer, to occur somewhere over land, is about 60 times smaller than the estimated 1000 years interval for such an event to occur at the specific location that it did. This gives a very rough estimate of a return period around 15 years with a lower bound of 1.5 years (not shown) to experience such a heatwave somewhere on land. A heatwave of this extent and extremity might therefore no longer be considered very rare if it could be expected anywhere around the globe in one or two decades. Further research on this and other exceptional heatwaves is needed to determine whether this estimate is indeed realistic, i.e., whether or not we should reject the assumption that this heatwave occurred by chance at this location."*

The quality of many figures is not good enough:

- Several figures are direct output of a climate explorer tool with a poor choice of colours, and confusing axis labels. For example, what is "max Tmax"? Tmax is used also in the text, however it is not defined. It took me a while to find out the difference between TXx and Tmax. It is very confusing and it is not explained.

*Thank you for pointing to the confusing mentions of 'max Tmax', which is indeed TXx. We change this in the figures and text. Furthermore, we improved the figures by deleting confusing titles and changing the confusing axis labels. Besides that we defined Tmax in the text.*

Fig. 12 and Fig 13 are not self-explanatory. The meaning of the colours is explained in the text, but not in the figure caption.

*We now also explain the colours in the figure caption: "The blue bars show ERA5 results, the light red bars the model results, with common model uncertainty shown as white bars around them. The model average is shown by the bright red bars. The magenta bars are the synthesized values, and the white box accompanying the magenta bars indicate the more conservative estimate of an unweighted average of observations and models."*

- Fig 12 shows the synthesis results of the **past** climate compared with **today's** climate. Why is "historical-ssp" written in the figure legend?

*We have written the additional "historical-ssp" and similar names in the legend, as well as the number of ensemble members between brackets. Then the reader can check whether a model result might be an outlier because of the model experiment or number of ensemble members. Note that the SSP scenario's have been used for the most recent years as well as for the future years. We now added to the figure caption:*

*"Model names include the model experiment and scenario and the number of ensemble members in brackets."*

- Fig. 15: I do not see the circulation changes described in the text. Assuming that the study area is the same as defined at the beginning of the manuscript, I cannot see at all a "southerly to southwesterly near surface flow" in the study area in panel B.

*We thank the reviewer for catching this. Figure 15 in the preprint draft didn't show 12UTC on the 29th of June 2021. Figure 15 has been updated to show 12UTC on the 29th of June 2021 and other times that more clearly show the dynamics at play and the southerly to southwesterly near-surface flow that advected marine air into the southwestern portion of the study area, causing cooling. Figure 15 also now includes the study area for further clarity. The figure caption has been updated accordingly, and text has been updated to read: "By 12UTC on the 29th of June 2021, the southwestern portion of the study area was under the influence of southerly to southwesterly near-surface flows that advected marine air and forced marked cooling."*

- Fig. 17: The figure label contains: "Fit: KNMI Climate Explorer". What is fitted in this case?

*The reviewer is right, this is not a fit, rather the origin of the map is the KNMI Climate Explorer. We will change this in the caption.*

- In case of the maps, it would be good to mark the study area.

*We added the study area to the figures where we think it is helpful (Figs 5b, 15 and 17).*

The structure of the paper is confusing:

- There are many sections and some of them are very short. For example, section 5 has 4 lines. It could be incorporated in section 6.

*We restructured some sections, and incorporated the model analysis section into the model validation section, renaming it to "Model evaluation and analysis"*

- The paper has no summary joining the main results from the different sections. The different topics (for example, the "statistical estimation", "synthesis results", "meteorological analysis") are like puzzle pieces which are not merged at the end.

*While working through the manuscript we clarified how different sections are related. For instance, we added introductory sentences in the synthesis section which summarise the findings of the paper up to that point, and made the first sentence of section "The broader context of the heatwave" more explicit in what is covered in the section. In the concluding section we reworded the first paragraph and joined the main results more specifically:*

*"This record-breaking extreme event has been analysed under the assumption that it was simply a low probability random event. A rudimentary calculation looking into the probability of a random event of this scale and severity to occur anywhere over the earth's land area, gave an estimated chance of order 1-in-15 years, which at first impression makes a random event seem plausible, but this should be more thoroughly investigated.*

*The alternative is that nonlinear interactions and feedbacks occurred, which amplified the intensity of this extreme, placing it in a different population of heatwave events with different (and possibly unknown) statistics. We briefly considered dynamical and hydrological (drought) mechanisms and modes of natural variability that could have had an amplifying role. The latter was found to have no important role in this event. The circulation itself was highly anomalous but not exceptional enough to explain the record breaking heat alone, however, local topography and preceding dryness may have amplified the associated temperatures. In addition, there may also be dynamical mechanisms (Arctic Amplification) at work that influence the persistence of blocking conditions.*

*Further research is planned to investigate whether these or other feedbacks were operating in this outstanding event, and whether those feedbacks are related to human-induced climate change and if they increase the frequency beyond that expected for random events of such extreme temperatures. Also, further research is needed into the limitations of standard GEV analysis on annual maxima with short records and seemingly non-stationary behavior.*

*Whether or not local or dynamical feedbacks are responsible for amplifying the extreme temperatures in this particular event, this study shows that the human-induced warming that has occurred since pre-industrial conditions does make extreme events like this possible in the current climate and study region, and many times more likely than in the pre-industrial era."*

- Line 136: "As discussed in section 1.2, we analyse the annual maximum of daily maximum temperatures (TXx)". This is discussed in section 1.1, actually.

*Thank you for spotting this, we changed it.*

- You mention all those different experiments in Sec. 2.2. The CM6A ensemble "is used to explore the influence of climate variability", you use high resolution GFDL and AMIP runs, but you do not explain what we learn from these runs, you just include them, together with all the other models, in the synthesis analysis.

*We used all models that were readily available, and with as many different experiments (model set-ups, forcings, grid size etc) as possible, to span a wide range of possible model outcomes and to avoid possible model errors that stem from a single model setting. Therefore it is useful that we have a couple of different model settings. Different results could stem from the difference in model experiments as well as simply from the fact that models are different. We do however not simply use all models in the synthesis: we only use those models that passed our validation tests.*

*In the synthesis figures the GFDL AMIP results are not distinctly different from the CMIP6 results. We now added to the synthesis section: "Generally, we do not see any consistent departures in the model results that can be traced back to experiment differences, except that models which consist of many ensemble members have smaller uncertainties."*

- The second part of the abstract is not clear and precise enough.

*We improved the abstract:*

*"Towards the end of June 2021, temperature records were broken by several degrees Celsius in several cities in the Pacific northwest areas of the U.S. and Canada, leading to spikes in sudden deaths, and sharp increases in emergency calls and hospital visits for heat-related illnesses. Here we present a multi-model, multi-method attribution analysis to investigate to what extent human-induced climate change has influenced the probability and intensity of extreme heatwaves in this region. Based on observations, modeling and a classical statistical approach the occurrence of a heatwave defined as the maximum daily temperatures (TXx) observed in the area 45 °N-52 °N, 119 °W-123 °W, was*

*found to be virtually impossible without human-caused climate change. The observed temperatures were so extreme that they lie far outside the range of historical temperature observations. This makes it hard to quantify with confidence how rare the event was.*

*Using a statistical analysis that assumes that the heatwave is part of the same distribution as previous heatwaves, i.e. it can be included in the statistical fit and it was not intensified by new nonlinearities, a first order estimation of the event frequency is of the order of once in 1000 years under current climate conditions. Using this assumption and in combining the results from the analysis of climate models and weather observations, we find that such a heat event would be at least 150 times less common without human-induced climate change. Also, this heatwave was about 2 °C hotter than a 1 in 1000-year heatwave would have been at the beginning of the industrial revolution, when global mean temperatures were 1.2 °C cooler than today. Looking into the future, in a world with 2 °C of global warming (0.8 °C warmer than today), a 1000-year event would be another degree hotter.*

*Our results provide a strong warning: our rapidly warming climate is bringing us into uncharted territory with significant consequences for health, well-being, and livelihoods. Adaptation and mitigation are urgently needed to prepare societies for a very different future."*

- The probability ratio PR is not defined.

*We added the definition where PR is mentioned for the first time:*

*"For the event under investigation we calculate the return periods, probability ratio (PR) and change in intensity as a function of GMST, where PR is defined as $PR = p_1/p_0$, with $p_1$ the probability of an event as strong as or stronger than the extreme event in the current climate and $p_0$ the probability in a counterfactual climate without anthropogenic emissions."*

---

## Referee Report (RR1)

I appreciate the effort of the authors in answering my comments / criticism and for implementing many of them in the new version of the manuscript. It think that the quality of the paper improved, and especially, the results are not just reported, but also explained and interpreted in the paper in a more critical way. I understand that it is difficult to perform attribution studies due to the limited amount of observations and different model biases, however this does not exclude a critical interpretation and a qualitatively high presentation of the results in the manuscript. I do not question that the message of the paper is very relevant and that action from authorities is urgently needed in order to prepare for the increasing frequency of heat events.

After reading the response of the authors and the new version of the manuscript, I still have a few comments / suggestions:

1) I think that the sentence "*it could either occur by chance or nonlinear effects have made such heatwave possible*" (L. 280-281) is an oversimplification leading to misunderstandings, thus it needs some clarification should be rephrased.
   We often model extreme events as random variables, but the climate is essentially a chaotic deterministic system. It behaves at certain scales as if it would be random (this is why the approximations using random variables often works), nonetheless extreme events do not happen by chance. What the authors actually mean, I suppose, is that we observe "by chance" a very low-probability event in a relatively short time series. This is explained at the beginning of Sec. 3, but this and similar sentences still appear in the manuscript.
   I am also not convinced that the only alternative to an event "by chance" are "nonlinear effects" or, as mentioned in the abstract, "new nonlinearities". It is possible that global warming makes some well-known processes more (or less) possible thus changing the distribution of extreme events. "New nonlinearities" sounds for me too specific considering that the authors did not study this issue directly.

2) L. 439-440 and L. 210-211 "*Also further research is needed into the limitations of standard GEV analysis on annual maxima with short records and seemingly non-stationary behavior*."
   The GEV approach is formulated for independent, identically distributed, random variables. It can be still applied to correlated data, in case the correlations are weak enough and the block maxima are uncorrelated. However, it is not surprising that it does not work if these conditions are strongly violated due to the non-stationarity induced by global warming. This issue is often addressed by assuming a time dependence of GEV parameters, however there is no theoretical support for these kind of dependencies. It is well known as well that the method is quite data-hungry because it considers only the maximum of each block. Furthermore, it is an asymptotic method, thus it is valid only in case the block size is large enough, and the convergence to the asymptotic distribution can be extremely slow. I do not question the advantages of the method, but the above mentioned application issues are well known in case the data does not satisfy the necessary conditions. Thus, care is needed when applying this method to observational data sets and non-stationary model simulations. But, again, these problems are well-know, thus I do not see the usefulness of the mentioned future studies.

   These issues are thoroughly explained in:
   Coles, "Introduction to Statistical Modeling of Extreme Values", *Springer*, New York, NY, USA, 2001.

For convergence issues and the problem of limited data size see, for example:

Vannitsem, "Statistical properties of the temperature maxima in an intermediate order Quasi-Geostrophic model," *Tellus, Series A: Dynamic Meteorology and Oceanography*, vol. 59, no. 1, pp. 80–95, 2007.

Felici, Lucarini, Speranza, and Vitolo, "Extreme value statistics of the total energy in an intermediate-complexity model of the midlatitude atmospheric jet. Part I: Stationary case", *Journal of the Atmospheric Sciences*, vol. 64, no. 7, pp. 2137–2158, 2007.

Gálfi, Bódai and Lucarini, Convergence of extreme value statistics in a two-layer quasi-geostrophic atmospheric model, *Complexity*, 5340858, 2017

3) The vast number of data sets and methods used in attribution studies seem to lead to shortcomings in the presentation of these methods and the interpretation of the results. If this could be avoided, however, it would make attributions studies more accessible for a broader audience and would reduce the risk of misinterpretation.
I am aware that there is a vast body of relevant scientific literature and the authors cannot explain every detail of their methods in the manuscript. However, I firmly believe that the relevant methods should be properly explained, even if very concisely. It is not reader friendly at all to give a list of papers and expect the reader to go through a whole body of literature just to understand one manuscript. Nonetheless, I think that also from this aspect the manuscript improved with respect to the previous version.

4) *Fig. 15* is too small, it is impossible to see the arrows illustrating the wind direction.

5) The language of the manuscript improved as well, but it still needs some revisions in terms of understandability and typos.

---

## Author Response (AR2)

**Reply to the reviewer**

I appreciate the effort of the authors in answering my comments / criticism and for implementing many of them in the new version of the manuscript. It think that the quality of the paper improved, and especially, the results are not just reported, but also explained and interpreted in the paper in a more critical way. I understand that it is difficult to perform attribution studies due to the limited amount of observations and different model biases, however this does not exclude a critical interpretation and a qualitatively high presentation of the results in the manuscript. I do not question that the message of the paper is very relevant and that action from authorities is urgently needed in order to prepare for the increasing frequency of heat events.

After reading the response of the authors and the new version of the manuscript, I still have a few comments / suggestions:

1) I think that the sentence "*it could either occur by chance or nonlinear effects have made such heatwave possible*" (L. 280-281) is an oversimplification leading to misunderstandings, thus it needs some clarification should be rephrased.
We often model extreme events as random variables, but the climate is essentially a chaotic deterministic system. It behaves at certain scales as if it would be random (this is why the approximations using random variables often works), nonetheless extreme events do not happen by chance. What the authors actually mean, I suppose, is that we observe "by chance" a very low-probability event in a relatively short time series. This is explained at the beginning of Sec. 3, but this and similar sentences still appear in the manuscript.
I am also not convinced that the only alternative to an event "by chance" are "nonlinear effects" or, as mentioned in the abstract, "new nonlinearities". It is possible that global warming makes some well-known processes more (or less) possible thus changing the distribution of extreme events. "New nonlinearities" sounds for me too specific considering that the authors did not study this issue directly.

We would like to thank the reviewer for pointing to the possible misinterpretation of these terms, as what the reviewer explains is exactly what we meant. We partly already explained it in the first paragraph of Sect **Observational analysis: return time and trend.** We now also change:

"With this approach we still assume this was an event happening by chance." into "With this approach we still assume this was an event happening by chance, that is, the behaviour is in line with that of a chaotic deterministic system in a warming climate and by chance we observe a low-probability event in this short time series."

In Sect **Probability of a chance event** we changed

"it could either occur by chance or nonlinear effects could have made such a heatwave possible" into

"it could either occur by chance (a low-probability event) or, for instance, nonlinear effects that have not been observed at this location before could have made such a heatwave possible"

**And in the abstract:**

new nonlinearities -> nonlinear interactions and feedbacks.

2)  L. 439-440 and L. 210-211"*Also further research is needed into the limitations of standard GEV analysis on annual maxima with short records and seemingly non-stationary behavior.*"
The GEV approach is formulated for independent, identically distributed, random variables. It can be still applied to correlated data, in case the correlations are weak enough and the block maxima are uncorrelated. However, it is not surprising that it does not work if these conditions are strongly violated due to the non-stationarity induced by global warming. This issue is often addressed by assuming a time dependence of GEV parameters, however there is no theoretical support for these kind of dependencies. It is well known as well that the method is quite data- hungry because it considers only the maximum of each block. Furthermore, it is an asymptotic method, thus it is valid only in case the block size is large enough, and the convergence to the asymptotic distribution can be extremely slow. I do not question the advantages of the method, but the above mentioned application issues are well known in case the data does not satisfy the necessary conditions. Thus, care is needed when applying this method to observational data sets and non-stationary model simulations. But, again, these problems are well-know, thus I do not see the usefulness of the mentioned future studies.
These issues are thoroughly explained in:
Coles, "Introduction to Statistical Modeling of Extreme Values", *Springer*, New York, NY, USA, 2001.

For convergence issues and the problem of limited data size see, for example:

Vannitsem, "Statistical properties of the temperature maxima in an intermediate order Quasi-Geostrophic model," *Tellus, Series A: Dynamic Meteorology and Oceanography*, vol. 59, no.

1, pp. 80–95, 2007.

Felici, Lucarini, Speranza, and Vitolo, "Extreme value statistics of the total energy in an intermediate-complexity model of the midlatitude atmospheric jet. Part I: Stationary case", *Journal of the Atmospheric Sciences*, vol. 64, no. 7, pp. 2137–2158, 2007.

Gálfi, Bódai and Lucarini, Convergence of extreme value statistics in a two-layer quasi-geostrophic atmospheric model, *Complexity*, 5340858, 2017

We changed the sentence "*Also, further research is needed into the limitations of standard GEV analysis on annual maxima with short records and seemingly non-stationary behavior.*" into "*Also, further research is needed on how to overcome the known limitations of standard GEV analysis on annual maxima with short records and very extreme values.*" as this better describes what the intention of the further research is, and acknowledges that the limitations are already known.

Note that we do not think that the challenges we encountered with the 2021 data point are caused by the general inappropriateness to fit a GEV to the TXx temperature data in this region, as the fit with GMST-dependence introduced works well to describe all data up to but excluding this event (Fig. 6). The difficulties encountered here are instead probably linked to shortness of the time series and the extremity of the event, or in addition, mechanisms previously not encountered in this region coming into play.

3)  The vast number of data sets and methods used in attribution studies seem to lead to shortcomings in the presentation of these methods and the interpretation of the results. If this could be avoided, however, it would make attributions studies more accessible for a broader audience and would reduce the risk of misinterpretation.
I am aware that there is a vast body of relevant scientific literature and the authors cannot explain

every detail of their methods in the manuscript. However, I firmly believe that the relevant methods should be properly explained, even if very concisely. It is not reader friendly at all to give a list of papers and expect the reader to go through a whole body of literature just to understand one manuscript. Nonetheless, I think that also from this aspect the manuscript improved with respect to the previous version.

We are pleased that the reviewer noticed an improvement with respect to the previous version. In this version, we added further detail in the statistical methods and synthesis sections to expand on our method.

In the statistical methods section we clarified some text and also add:

Uncertainties corresponding to the statistical-model uncertainty, are obtained using a non-parametric bootstrap procedure. With this GEV distribution, first the PR and intensity change are calculated from observations, as well as the return period in the current climate. Next, the return period is used as a threshold to specify the event magnitude for the models. For this return period, the PRs and intensity changes between 2021 and the counterfactual climate are calculated from different models. This is, however, only done for models that pass our validation tests on the seasonal cycle, the spatial pattern of the climatology, and the scale and shape parameters of the GEV distribution, see Section 4. Finally, both observational and model results are synthesised into a consistent attribution statement, see Section 5.

In the synthesis section we clarified some text and also add:

"The uncertainty due to differences in model set up and physics is represented by model spread --- the average departure of each model from the mean model best estimate. This is added in quadrature to the model natural variability as white extensions to the light red bars in the synthesis figures. The uncertainty in the model average (bright red bar) consists of a weighted mean uncertainty, where the contribution from each model is inversely proportional to the uncertainty due to natural variability squared, plus the model spread term added in quadrature to the uncertainty in the weighted mean. Please see e.g. Kew et al. (2021) for more detailed information on the synthesis technique including how weighting is calculated for models."

Note that we intentionally use many different data sets and methods to increase the robustness of the results, i.e. we aim to capture something of the method-related uncertainty, represented by the intermodel spread. The synthesis figures show the results from each method as well as the synthesised combination, so it can be clearly seen if there are outliers. We hope that the general message - that warm extremes in this region are becoming more frequent and more extreme and action should be taken - is accessible to a broader audience. We are also aware that some of the challenges we encountered are being investigated in other studies.

4) *Fig. 15* is too small, it is impossible to see the arrows illustrating the wind direction.

The figure was indeed intended to cover the full width of the page, which was not true in the manuscript. We also enlarged the arrow size so that the arrows are more visible.

5) The language of the manuscript improved as well, but it still needs some revisions in terms of understandability and typos.

We have improved readability and corrected typos as well as unclear sentences.